



# Properties of black carbon and other insoluble light-absorbing particles in seasonal snow of northwest China

5    Wei Pu[1], Xin Wang[1], Hailun Wei[1], Yue Zhou[1], Jinsen Shi[1], Zhiyuan Hu[1], Hongchun Jin[1],

Quanliang Chen[2]

[1] Key Laboratory for Semi-Arid Climate Change of the Ministry of Education, College of Atmospheric Sciences, Lanzhou University, Lanzhou, 730000, China

10  [2] College of Atmospheric Science, Chengdu University of Information Technology, and Plateau Atmospheric and Environment Laboratory of Sichuan Province, Chengdu 610225, China

Correspondence to: Xin Wang (wxin@lzu.edu.cn)

Submitted to The Cryosphere





**Abstract.** A large field campaign was conducted in northwestern China from January to February 2012 to measure the insoluble light-absorbing particles (ILAPs) and chemical species in the snow, and two hundred eighty-four snow samples were collected at 38 sites in Xinjiang Province and 6 sites in Qinghai Province. The

cleanest snow was found in northeastern Xinjiang along the border of China, and it presented an estimated black carbon (BC) ($C_{BC}^{est}$) of approximately 5 ng g$^{-1}$. The dirtiest snow was found near industrial cities, and it presented a $C_{BC}^{est}$ of approximately 450 ng g$^{-1}$. Overall, the $C_{BC}^{est}$ of most of the snow samples collected in this campaign was 10-150 ng g$^{-1}$. Vertical variations in the snowpack ILAPs indicated

a probable shift in emission sources with the progression of winter. An analysis of the fractional contributions to absorption implied that organic carbon (OC) dominated the 450-nm absorption in Qinghai, whereas the contributions from BC and OC were comparable in Xinjiang. A Positive Matrix Factorization (PMF) model was run to explore the sources of particulate light absorption, and the results indicated an optimal

3-factor/source solution that included industrial pollution, biomass burning, and soil dust. In addition, the chemical components were evaluated to examine the mass contributions. In Qinghai, biomass burning was the dominant absorption factor despite the high mass contribution from soil dust. In Xinjiang, the primary absorption source was industrial pollution at sites near cities and biomass burning at most sites in

other regions. A negative correlation was observed between the BC mixing ratio and altitude in Xinjiang. An analysis based on the PMF 3-factor solution showed that this relationship likely resulted from gradient variations in the contributions of industrial



pollution sources.

# 1 Introduction

The deposition of insoluble light-absorbing particles (ILAPs), primarily black carbon

(BC), organic carbon (OC), and dust, on snow can reduce snow albedo (Warren and

Wiscombe, 1980; Chylek et al., 1983; Brandt et al., 2011; Hadley and Kirchstetter,

2012), which can significantly affect regional and global climate (Jacobson, 2002,

2004; Hansen and Nazarenko, 2004; Flanner et al., 2007, 2009; McConnell et al.,

2007; Ramanathan and Carmichael, 2008; Bond et al., 2013). Warren and Wiscombe

(1980) suggested that a mixing ratio of 10 ng $g^{-1}$ of BC in snow may reduce the snow

albedo by approximately 1%. A modeling study indicated that soot could reduce snow

and sea ice albedo by 0.4% from the global average and by 1% in the Northern

Hemisphere (Jacobson, 2004). Previous studies found that the "efficacy" of BC/snow

forcing in the Arctic is more than three times greater than that of forcing by $CO_2$

(Hansen and Nazarenko, 2004; Flanner et al., 2007). However, radiative forcing is

highly uncertain. For example, Hansen and Nazarenko (2004) found that the effect of

soot on snow and ice albedo yielded a climate forcing of +0.3 W $m^{-2}$ in the Northern

Hemisphere. Recently, the IPCC's AR5 (2013) reported that the radiative forcing from

BC in snow and ice is 0.04 W $m^{-2}$ of the global mean, although it presents a low

confidence level. Bond et al. (2013) indicated that the best estimate of climate forcing

from BC deposition on snow and sea ice in the industrial era is +0.13 W $m^{-2}$ with 90%

uncertainty bounds of +0.04 to +0.33 W $m^{-2}$. The all-source present-day climate





forcing including preindustrial emissions is somewhat higher at +0.16 W m$^{-2}$. Many

factors complicate the evaluation of climate forcing by BC in snow (Hansen and

Nazarenko, 2004; Bond et al., 2013). Hence, abundant comprehensive field

campaigns are required to measure ILAPs in snow to limit this uncertainty.

Recently, a number of field campaigns have been conducted to measure the BC in

snow and ice in the Arctic (Clarke and Noone, 1985; Chylek et al., 1987, 1995;

Cachier and Pertuisot, 1994; Grenfell et al., 2002; Hagler et al., 2007a, 2007b;

McConnell et al., 2007; Forsstrom et al., 2009; Doherty et al., 2010, 2013). Overall,

the BC mixing ratios in the snow of the Arctic were 3-30 ng g$^{-1}$. The cleanest snow

was found on the Greenland Ice Sheet, whereas the dirtiest snow was found in East

Russia (Doherty et al., 2010). Modeling studies have also evaluated the effect of BC

in snow on warming in the Arctic (Hansen and Nazarenko, 2004; Jacobson, 2004;

Flanner et al., 2007, 2009; Koch et al., 2009; Shindell and Faluvegi, 2009). Hansen

and Nazarenko (2004) suggested that the mean effect of soot on the spectrally

integrated albedos in the Arctic is 1.5%. Flanner et al. (2007) noted that the simulated

annual Arctic warming is 1.61 ℃ and 0.50 ℃ for 1998 and 2001 central experiments

when BC is included in snow compared with control simulations without BC.

However, limited field campaigns and modeling studies are available in North

America (Chylek et al., 1987; Qian et al., 2009; Hadley et al., 2010; Dang and Hegg,

2014; Doherty et al., 2014), Europe (Eleftheriadis et al., 2009; Thevenon et al., 2009;

Painter et al., 2013; Gabbi et al., 2015; Peltoniemi et al., 2015), the Tibetan Plateau

(Xu et al., 2006, 2009, 2012; Kang et al., 2007; Ming et al., 2008, 2009; Qiu, 2008;

Cong et al., 2013; Wang et al., 2014; Li et al., 2016), and North China (Huang et al., 2011; Ye et al., 2012; Wang et al., 2013; Zhao et al., 2014), where snow is closer to the sources of ILAPs and more exposed to sunlight; therefore, the effect of ILAPs on snow may be more significant. Hence, we conducted a large field campaign across

northwestern China from January to February 2012 to measure the ILAPs in snow.

In addition to BC, which presented the most absorptive impurity per unit mass, OC and dust can significantly contribute to particulate light absorption in snow. OC in snow may be related to either combustion products that are deposited onto snow or soil that is mixed into snow. Xu et al. (2006) first quantified the OC content on the

Tibetan Plateau and determined the effect of OC on surface snow melting. Wang et al. (2013) suggested that OC dominates the particulate light absorption across the grasslands of Inner Mongolia in North China. Light absorption by dust is usually related to iron oxides. Although its ability to reduce snow albedo is less than that of BC by approximately a factor of 50 (Warren, 1984), dust is the dominant absorber in

snow locations. For example, the increased radiative forcing by dust in snow has affected the timing and magnitude of runoff from the Upper Colorado River Basin (Painter et al., 2007, 2010).

Understanding the sources of ILAPs in snow is necessary for examining the climatic effects of ILAPs in snow. Certain scientists have focused on exploring the potential

sources of BC in snow (Flanner et al., 2007; Shindell et al., 2008; Forsstrom et al., 2009); however, these studies primarily relied on numerical transport modeling based on limited data from emission inventories or calculated back trajectories, and they

showed a limited ability to quantify the source attribution of BC. Recently, a standard

receptor model has been successfully used to resolve the sources of snow BC. For

example, Hegg et al. (2009, 2010) utilized measured ILAPs and chemical components

as inputs to run the model. The results showed four sources (crop and grass burning,

boreal biomass burning, pollution, and marine) of light-absorbing particles in the

Arctic snow. Biomass burning sources dominated during spring, although pollution

played a more significant role during fall and winter and in summer in Greenland.

Zhang et al. (2013a) evaluated the source attribution of ILAPs in the snow in

northeastern China and found three sources for the ILAPs in the snow pack: soil dust

(53%), industrial/urban pollution (27%), and biomass and biofuel burning (20%).

These authors concluded that soil dust was the dominant source in the northeastern

area of China, which is consistent with assessments that were based on back trajectory

cluster analyses. A similar study by Doherty et al. (2014) focused on the sources of

snow BC in North America and found that both biomass and fossil fuel combustion

were the main sources of snow BC in the Pacific Northwest, Intermountain West and

Canada, whereas soil dust played a predominant role in particulate light absorption in

the northern U.S. Plains. Obviously, analyses based on the receptor model are timely

and useful for identifying the source attribution of snow BC; thus, this model can be

used to assess the potential sources of ILAPs in the snow in northwestern China

because of the variety of emission sources (Ye et al., 2012).

In this study, we analyzed the spatial and vertical distributions of ILAPs in the

seasonal snow in northwestern China, investigated the contributions from BC, OC and

Fe to particulate light absorption, and estimated the source attributions of snow BC.

Finally, we evaluated the chemical components to examine the potential emission

sources.

**2 Methods**

**2.1 Snow collection**

Seasonal snow was collected at 38 sites in Xinjiang and 6 sites in Qinghai in China

from January to February 2012, with 284 snow samples obtained across the entire

expedition. Figure 1 shows the locations of the sampling sites, which were numbered

in chronological order and followed the field campaign by Wang et al. (2013). Fresh

snow was gathered from 13 sites where snow was falling at the time of sampling.

Forty-two sites were separated into 5 regions according to their geographical

distribution to investigate the spatial variations in snow-containing contaminants and

their potential sources, with one region (Region 1) located in Qinghai and the other

regions (Regions 2-5) located in Xinjiang (Figure 1).

Normally, snow was collected at vertical intervals of 5 cm from the top to the bottom

at each site. If distinct layering was present, such as a melt layer or a top layer of

newly fallen snow, that layer was sampled individually. In Qinghai, the snow was thin

and patchy at sites 47-49; therefore, the samples were collected from drift snow. Left

and right snow samples were gathered within each layer in two adjacent vertical

profiles to compare and average the sample pairs. The snow density and temperature

were also measured within each layer to quantify the deposition flux of the BC or

other ILAPs. The sampling sites were selected 50 km from cities and at least 1 km


upwind of the approach road or railway to minimize the effect of pollution from local

sources and achieve a representation of large areas.

The snow samples were filtered at four temporary laboratories to prevent the melting

snow from influencing the ILAP content. The snow samples were quickly melted in a

5 microwave oven and then immediately filtered through a 0.4-μm Nuclepore filter. The

samples "before" and "after" filtration were collected and refrozen for subsequent

chemical analyses, and the filters were subjected to BC and OC analyses. Additional

details on the snow collection and filtration processes have been previously reported

(Doherty et al. 2010, 2014; Wang et al., 2013).

**2.2 Chemical speciation**

The chemical analysis performed here followed that of Wang et al. (2015) and was

similar to the procedures described by Zhang et al. (2013a) and Doherty et al. (2014).

Details of these approaches have been previously reported (Yesubabu et al., 2014).

Briefly, the major ions ($SO_4^{2-}$, $NO_3^-$, $Cl^-$, $F^-$, $Na^+$, $K^+$, $Ca^{2+}$, $Mg^{2+}$ and $NH_4^+$) were

15 analyzed with an ion chromatograph (Dionex, Sunnyvale, CA, USA), and the trace

elements (e.g., Fe, Al, Cu, Mn, Cr, and Ba) were measured by inductively coupled

plasma mass spectrometry (ICP-MS). Pairs of unfiltered and filtered snow water

samples were analyzed to evaluate the possible effect of filtering on the chemical

constituents, and obvious differences were not observed. Mineral dust (MD), Cl salt,

biosmoke K ($K_{Biosmoke}$) and trace element oxides (TEO) were determined to assess the

mass contributions of the major components in the surface snow. The mineral dust

content was calculated by a straightforward method, and the Al concentration in dust



was estimated at 7% (Zhang et al., 2013b):

$$MD=Al/0.07 \tag{1}$$

Cl salt was estimated as follows in accordance with Pio et al. (2007) and Zhang et al. (2013b):

$$Cl\ salt=Na_{Ss}^{+}+Cl^{-}+Mg_{Ss}^{2+}+Ca_{Ss}^{2+}+K_{Ss}^{+}+SO_{4Ss}^{2-}$$

$$=Na_{Ss}^{+}+Cl^{-}+0.12Na_{Ss}^{+}+0.038Na_{Ss}^{+}+0.038Na_{Ss}^{+}+0.25Na_{Ss}^{+} \tag{2}$$

where $Na_{Ss}^{+}$ is sea salt $Na^{+}$, 0.12, 0.038, 0.038, and 0.25 are the mass ratios in seawater of magnesium to sodium, calcium to sodium, potassium to sodium and sulfate to sodium, respectively. $Na_{Ss}^{+}$ is calculated as follows (Hsu et al., 2009):

$$Na_{Ss}=Na_{Total}-Al*(Na/Al)_{Crust} \tag{3}$$

where $(Na/Al)_{Crust}$ is the Na/Al ratio of representative dust materials (Wedepohl, 1995). Following Hsu et al. (2009), we estimated all three fractions (dust, sea salt, and biosmoke fractions) of K in snow, and $K_{Biosmoke}$ was determined as follows:

$$K_{Biosmoke}=K_{Total}-K_{Dust}-K_{Ss} \tag{4}$$

$$K_{Dust}=Al*(K/Al)_{Crust} \tag{5}$$

$$K_{Ss}=Na_{Ss}*0.038 \tag{6}$$

where $(K/Al)_{Crust}$ is 0.37 and represents the K/Al ratio in the dust materials (Wedepohl, 1995) and $Na_{Ss}$ is estimated by Equation (3). Following Zhang et al. (2013b), we calculated the contribution of TEO using the following equation:

$$TEO=1.3*\sum_{i}(\alpha_{i}*TE_{i}) \tag{7}$$

where TE represents the trace elements (Ti, V, Cr, Mn, Co, Ni, Cu, Zn, As, Se, Sr, Mo, Ag, Cd, Sn, Ba, Hg, Tl, Pb, Th, U, and Be) determined via ICP-MS and α is the



weight coefficient of each trace element. $\alpha$ is a function of the enrichment factor (EF):

$\alpha=0$,      $EF \leq 1$;

$\alpha=0.5$,   $1 < EF < 5$;                                       (8)

$\alpha=1$,      $EF \geq 5$.

where the EF of a given element (E) is calculated by the equation $EF=(E/Al)_{Snow}/(E/Al)_{Crust}$ (Hsu et al., 2010). $(E/Al)_{Snow}$ and $(E/Al)_{Crust}$ are the ratios of the elements to the Al concentration in the snow sample and crust (Wedepohl, 1995), respectively. A multiplicative factor of 1.3 was used to convert the

element abundance to the oxide abundance, which is similar to the method of Landis et al. (2001).

### 2.3 Spectrophotometric analysis

The filters were analyzed for the ILAP content in the snow using a modified integrating-sandwich spectrophotometer (ISSW), which was described by Grenfell et

al. (2011) and used by Doherty et al. (2010, 2013, 2014, 2015), Dang and Hegg (2014) and Wang et al. (2013, 2015). This ISSW spectrophotometer measures the light attenuation spectrum from 400 to 700 nm where the signal-to-noise ratio is optimized. The total light attenuation spectrum is extended over the full spectral range by linear extrapolation from 400 to 300 nm and from 700 to 750 nm (Grenfell et al., 2011).

Light attenuation is nominally only sensitive to ILAPs on the filter because of the diffuse radiation field and the sandwich structure of two integrated spheres in the ISSW (Doherty et al., 2014). By considering all of the light absorption that occurred

from 650-700 nm, the maximum possible BC ($C_{BC}^{max}$) mixing ratio was calculated by

calibrating the results against a set of fullerene (Alfa Aesar, Inc., Ward Hill, MA, USA)

standard filters with a mass absorbing coefficient (MAC) of 6.3 $m^2$ $g^{-1}$ at 550 nm. In

general, BC, OC and dust (Fe) dominated the light absorption by ILAPs in the snow

samples. The equivalent BC ($C_{BC}^{equiv}$), estimated BC ($C_{BC}^{est}$), absorption Ångström

exponent of all ILAPs ($Å_{tot}$), and light absorption fraction by non-BC ILAPs ($f_{nonBC}^{est}$)

were calculated by using the wavelength dependence of the measured spectral light

absorption and by assuming that the MACs of the BC, OC and Fe were 6.3, 0.3, and

0.9 $m^2$ $g^{-1}$ at 550 nm, respectively, and that the absorption Ångström exponents (Å)

were 1.1, 6, and 3, respectively. The details of this analysis were interpreted according

to Grenfell et al. (2011). The OC mixing ratio was also determined according to

Equation (2) in Wang et al. (2013), and the Fe concentration was determined

according to the ICP-MS measurements.

Many studies (e.g., Jacobson, 2001; Hadley and Kirchstetter, 2012; Bond et al., 2013)

have indicated that the MAC of BC is somewhat higher than the value used here, and

Bond and Bergstrom (2006) recommended a value of 7.5±1.2 $m^2$ $g^{-1}$ at 550 nm.

However, we applied a value of 6.3 $m^2$ $g^{-1}$ to provide a comparison with previous

studies (Hegg et al., 2009, 2010; Doherty et al., 2010; Wang et al., 2013). If the MAC

of BC is actually close to 7.5 $m^2$ $g^{-1}$, then our measured mass mixing ratio will be too

high by a factor of 1.19. If the radiation models are run with the BC mixing ratio

reported in this study, then the MAC of BC must be 6.3 $m^2$ $g^{-1}$; otherwise, the BC

mixing ratio should be scaled appropriately.



### 2.4 PMF model

The Positive Matrix Factorization (PMF) model that was used here (US EPA PMF 5.0) is a receptor model that can quantify contributions from sources to samples based on the composition or fingerprints of the sources, and it has been widely applied (e.g., Amato et al., 2009; Amato and Hopke, 2012). The speciation or composition is determined by using analytical methods appropriate for the media, and key species or combinations of species are required to distinguish the effects. The PMF model is a multivariate factor analysis tool that decomposes a matrix of speciated sample data into two matrices: factor contributions and factor profiles. These factor profiles must be interpreted by the analyst to identify the source types that may have contributed to the sample by using available ancillary information, such as the measured source profile information and emission or discharge inventories. The characteristic factor profiles are completely dependent on the mathematical approaches of the PMF model; therefore, the number of factors is not known a priori and must be selected individually in terms of the analyst's understanding of the sources that affect the samples as well as the number of samples and species' characteristics.

The PMF model uses two data sets as inputs to weigh individual points. One is the set of the concentrations of the input species, including the chemically analyzed constituents, along with $C_{BC}^{max}$, and the other is the associated uncertainty data set. Uncertainty estimates of the chemical concentrations are based on analyses of replicate standards with uncertainties that are calculated as twice the standard error of the mean for each analyzed species. The uncertainties of $C_{BC}^{max}$ were calculated as per


Doherty et al. (2010). $C_{BC}^{max}$ was likely biased by errors in the assumed MAC of our fullerene standards, which was applied to convert the measured absorption to a maximum BC mass. However, the relative biases of $C_{BC}^{max}$ were uniform across all of the data sets, and the results of the PMF analysis were dependent on the relative

variance of the given species against the absolute concentration; therefore, the effects can be appropriately ignored (Doherty et al., 2014).

Normally, the PMF model is applied to analyze a time series of species concentrations at a single observation site to estimate temporal variations. In this study, we used the PMF model to analyze the spatial variations in source contributions. Although

atypical, previous studies have effectively employed this model and confirmed its reliability in terms of factor analyses of spatial distributions (Paatero et al., 2003; Chen et al., 2007; Hegg et al., 2009, 2010; Zhang et al., 2013a; Doherty et al., 2014).

**3 Results**

Table 1 summarizes the 2012 northwestern China field campaign and lists the values

for the $C_{BC}^{equiv}$, $C_{BC}^{max}$, $C_{BC}^{est}$, $Å_{tot}$, and $f_{nonBC}^{est}$ at every sampling layer for each site. The cleanest snow was found in the far north of Xinjiang Province (sites 70, 75, 77 and 78) and the high-altitude sites in the Tianshan Mountains (sites 79 and 82) because these sites were located far from the emission sources; moreover, the melting-amplified effect on the BC mixing ratios was negligible because the field campaign was

conducted in January and February when the snow had not yet melted. Although the $C_{BC}^{est}$ result was low and reached values of 1 ng g$^{-1}$ at site 77, the $C_{BC}^{est}$ estimates were considered too uncertain when the corresponding $f_{nonBC}^{est}$ was > 85% according to



Doherty et al. (2014); therefore, we did not consider these values. Thus, the lowest

$C_{BC}^{est}$ values were approximately 5 ng $g^{-1}$, which were smaller than the minimum BC

mixing ratio of approximately 40 ng $g^{-1}$ measured in North China via the same

spectrophotometric analysis (ISSW) (Huang et al., 2011; Wang et al., 2013) and

comparable to the values of approximately 3 ng $g^{-1}$ from the Greenland Ice Sheet

(Doherty et al., 2010). The highest $C_{BC}^{est}$ was found at sites 53, 60, 67, 83 and 84. At

site 83, the $C_{BC}^{est}$ reached 619 ng $g^{-1}$ at the bottom layer; however, the underlying soil

may have been responsible for this high value. Therefore, this value cannot represent

the regional background level of BC. After excluding site 83, the highest $C_{BC}^{est}$ value

was approximately 450 ng $g^{-1}$, which was much lower than the values of >1000 ng

$g^{-1}$ in snow in the industrial area of northeastern China (Wang et al., 2013). Overall,

the $C_{BC}^{est}$ of most of the snow samples ranged from 10-150 ng $g^{-1}$, which were similar

to the visible values reported by Ye et al. (2012) and BC measurements of 4-120 ng

$g^{-1}$ recorded from glaciers in Tibet and Xinjiang by a previous field campaign that

used a controlled combustion method (Xu et al., 2006, 2009; Ming et al., 2008, 2009).

### 3.1 Results by region

As discussed above, the sample sites were separated into five regions. Table 2 lists the

regional averages and standard deviations of $C_{BC}^{equiv}$, $C_{BC}^{max}$, $C_{BC}^{est}$, $Å_{tot}$, and $f_{nonBC}^{est}$ for

the surface and subsurface layers. The spatial distributions of $C_{BC}^{est}$ and $Å_{tot}$ for the

surface snow samples are shown in Figure 2a and 2b. In addition, Figure 2c shows the

equivalent BC value used to explain the 300-750-nm absorption by all non-BC ILAPs,

which was calculated as $C_{BC}^{equiv} * f_{nonBC}^{est}$. Thus, Figures 2a and 2c can be compared to





determine the relative contributions from BC and non-BC ILAPs to the snow albedo

reduction at each site. However, the BC mass deposited onto snow over a specified

period is more useful when comparing models than the surface values because the

average BC over many snowfall events across a typical month or season presents a

more representative contribution to the background levels throughout the entire

accumulation period (Doherty et al., 2014). Thus, in Table 3, we list the integrated

snow water equivalent (SWE) and the total BC mass for a 1-cm$^2$ column of snow

(integrated BC). We also estimated the average BC mixing ratios in the snow column

($\overline{C_{BC}^{est}}$), which were calculated as the integrated BC divided by the SWE (Figure 2d and

Table 3). Indeed, the $\overline{C_{BC}^{est}}$ value was more spatially uniform than the $C_{BC}^{est}$ value.

In Region 1 (sites 47-52), which is located in the eastern Tibetan Plateau in Qinghai

Province, the snow was thin and patchy and presented a sample snow depth of 2.5-10

cm. During windy periods, local soil can be lofted and deposited onto snow, and this

deposition has been confirmed by previous reports (Ye et al., 2012), which observed

yellowed filters because of the heavy loading of soil dust. Although the $C_{BC}^{est}$ was

intermediate and presented typical values of 30-150 ng g$^{-1}$, both the $C_{BC}^{equiv}$ and

$C_{BC}^{equiv} * f_{nonBC}^{est}$ values were highest in the surface (307±119, 213±89 ng g$^{-1}$) and

subsurface (332±201, 214±116 ng g$^{-1}$) snow among all five regions (Table 2 and

Figure 2) when considering the primary proportion of non-BC ILAPs in the mass of

the ILAPs. In addition, the Å$_{tot}$ values were 3.6-4.6 and higher than those in any

other region, which indicated the predominant contribution from non-BC ILAPs to

light absorption.

Region 2 (sites 53-59, 61, and 79) is located on the Tianshan Mountains, and except

for site 58, all of the sites were located far from cities. Site 58, which is situated in the

foothills near Yili Prefecture, likely experiences a greater influence of human

activities. All of the regional statistics were intermediate. The $C_{BC}^{est}$ values were

generally in the range of 20-100 ng g$^{-1}$. At site 54, the surface $C_{BC}^{est}$ was low and

reached 7 ng g$^{-1}$. Higher altitudes (> 3000 m) and freshly fallen snow at the surface

may have been responsible for this lower value. The higher mixing ratios at site 53 for

both surface and subsurface snow may have been caused by local soil dust or biomass

burning as reported by Ye et al. (2012). The $Å_{tot}$ values varied remarkably from 2.2

to 4.5.

Region 3 (sites 60, 62-63, and 80-84) is located to the north of the Tianshan

Mountains, and industrial cities are located close to this region, and the population

density is much higher than that in any of the other regions; therefore, human

activities may have dominated the contributions to ILAPs in the snow. However, the

$C_{BC}^{est}$ values, which primarily ranged from 10-100 ng g$^{-1}$, were comparable to those in

other regions in Xinjiang, which were inconsistent with the findings in North China,

where the $C_{BC}^{est}$ in the industrial northeastern area was higher by a factor of 10

compared with the values to the remote northeast on the border of Siberia (Wang et al.,

2013). The $f_{nonBC}^{est}$ for the surface and subsurface samples was considerable different

and presented values of 58±15% and 72±12%, respectively, which reflects the

temporal changes in the contributions from BC and non-BC particles to light

absorption over time. The $Å_{tot}$ values were typically 2-4, which were comparable to



those in Region 2, despite the snow samples may have been more affected by human activities.

Region 4 (sites 64-70) is located in northwestern Xinjiang along the border of China. The $C_{BC}^{est}$ generally ranged from 20-150 ng g$^{-1}$. The regional average $Å_{tot}$ was approximately 3. The $f_{nonBC}^{est}$ in this region was the lowest and presented an average value near 50%; therefore, the BC and non-BC particles presented almost identical contributions to light absorption, which was inconsistent with the other regions, where non-BCs played a dominant role.

The cleanest snow of this campaign was found in Region 5 (sites 72-78) in northeastern Xinjiang. Most of the snow samples had low $C_{BC}^{est}$ values of 10-50 ng g$^{-1}$, which were much smaller than the values of 50-150 ng g$^{-1}$ in the cleanest snow in northeastern China. The $Å_{tot}$ value generally ranged from 2.5 to 3.5 and presented a regional average of approximately 3, which was consistent with other regions in Xinjiang. The $f_{nonBC}^{est}$ varied obviously varied from < 50% to > 90%. This wide range indicates the spatial variance in the dominant emission sources of particulate light absorption in this region.

### 3.2 Vertical variations in snowpack light-absorbing particulates

The vertical profiles of the $C_{BC}^{max}$, $C_{BC}^{est}$, and $Å_{tot}$ at each sample site are shown in Figure 3. At sites 49-52 in Qinghai (Region 1), an obvious dust layer was present in each site, and analyses of the ILAP content by ISSW presented high uncertainties. Therefore, we did not report the values at these layers. In addition, the sampling at sites 47-49 was conducted in drift snow as discussed above. Thus, the vertical profiles



from these sites in Qinghai did not accurately represent the temporal variations in the deposition of snow, although apparent vertical differences were observed at could sites, such as site 47. In Regions 3, 4 and 5 in Xinjiang, the $C_{BC}^{est}$ values were much larger in the surface snow (127±158, 126±124, and 74±56 ng g$^{-1}$, respectively) than in

the subsurface snow (75±120, 82±56, and 37±31 ng g$^{-1}$, respectively), with the ratio of the $C_{BC}^{est}$ from the top layer to the average $C_{BC}^{est}$ from all the subsurface layers presenting values of 1.7, 1.5, and 2, respectively, which indicates an increase in aerosol deposition later in winter. However, the $C_{BC}^{est}$ values in the surface (81±102 ng g$^{-1}$) and subsurface layers (89±69 ng g$^{-1}$) in Region 2 were comparable. These

differences in deposition could have been caused by changes in the dry deposition and/or the mixing ratio of the ILAPs to the precipitation water content in snowfall. Doherty et al. (2013) studied the observed vertical redistribution of BC and other ILAPs in melting snow and noted that melt amplification generally appears within the top few centimeters of the snowpack, and it increases the BC mixing ratios of the

surface snow. However, the high surface BC values reported here could not have originated from this mechanism because the temperature at these sites was not high enough to melt the snow. The $Å_{tot}$ and $f_{nonBC}^{est}$, the values increased by 0.41 and 10% from the subsurface to the surface snow in Region 2, respectively, but decreased by 0.40 and 14% in Region 3, respectively. These variations may have been partly caused

by a shift in emission sources as winter progressed, with a greater contribution to the ILAPs in snow from biomass burning in Region 2 and fossil fuel burning in Region 3 during this season.





### 3.3 Contributions to particulate light absorption by BC, OC and Fe

Figure 4a shows the fractional contributions of BC, OC and Fe to light absorption at
450 nm for the surface snow samples at each site. In Qinghai (Region 1), OC
dominated the absorption, and the fractional contribution reached approximately 70%.

However, dust accounted for the main portion of the particulate mass, which was
confirmed by the yellow color of the filters, and the fractional contribution from Fe to
absorption (only approximately 5%) was not significant. In Xinjiang, the fractional
contributions from BC, OC and Fe were approximately 45%, 50%, and 5%,
respectively, and their patterns did not vary greatly by region. At sites 58, 75, and 77,

the fraction of light absorption from Fe exceeded 20%. This value was much higher
than the values at the other sites, which may be explained by two possible reasons: (1)
less BC and OC were contributed by biomass and fossil fuel burning and more Fe was
contributed by dust, and (2) Fe was contributed by industrial emissions as well as dust.
Wang et al. (2013) analyzed the snow particulate light absorption by ILAPs in the

snow samples in northeastern China and noted that OC could be composed of soil
organics, combustion aerosols, and/or other biological organics. BC and OC could be
produced via combustion sources, such as fossil fuel burning, open burning and
biofuel burning. Thus, we could not precisely separate the contributions from different
sources by analyzing the ILAPs. Hence, a more feasible method is required to

evaluate and quantify the contributions from emission sources to particulate light
absorption (see Section 3.4).

The relative contributions of these three species are related to the values of $\text{Å}_{tot}$ and

$\text{Å}_{nonBC}$, which are shown in Figure 4b. Overall, the $\text{Å}_{nonBC}$ values were almost in a narrow range of 5-6, which indicated that OC was the major component of non-BC ILAPs because the Å values of OC and Fe were 6 and 3, respectively. Three exceptions were sites 58, 75, and 77, which had $\text{Å}_{nonBC}$ of $< 5$, and higher fractional

contributions from Fe to absorption likely resulted in these lower values. Compared with $\text{Å}_{nonBC}$, the $\text{Å}_{tot}$ values varied greatly from 1.84 to 4.66, and the large variety of BC mixing ratios was mainly responsible for the large changes in $\text{Å}_{tot}$. In terms of the spatial distribution, the $\text{Å}_{tot}$ value was close to the $\text{Å}_{nonBC}$ value in Qinghai (Region 1), although the differences between $\text{Å}_{tot}$ and $\text{Å}_{nonBC}$ were highly variable in

Xinjiang.

As discussed in Section 2.3, we attributed the snow particulate absorption to BC, OC, and Fe based on the absorption optical depth measured by ISSW along with the chemical analyses of Fe and assumptions for the Å values and mass absorbing coefficients of BC, OC, and Fe. Furthermore, $\text{Å}_{nonBC}$ was determined by the light

absorption of OC and Fe. However, the assumptions presented large uncertainties, which may have introduced considerable errors in the values of BC, OC, and $\text{Å}_{nonBC}$. Doherty et al. (2010) analyzed the errors that originated from these assumptions and indicated a likelihood of uncertainty of $< 50\%$ based on liberal evaluations of these potential sources of errors.

**3.4 PMF results**

**3.4.1 PMF source profile**

The PMF 5.0 model was run using the chemical and spectrophotometric analysis data

set of the surface snow samples. Following Doherty et al. (2014), we used $C_{BC}^{max}$ to

estimate the fractional contributions to the 650-700-nm particulate absorption by all of

the potential emission sources based on two reliable reasons: (1) $C_{BC}^{max}$ represents the

mass of BC, assuming all of the particulate light absorption (650-700 nm) is related to

BC; and (2) $C_{BC}^{max}$ is only calculated based on the assumed MAC of BC; therefore,

the errors of $C_{BC}^{max}$ were the lowest among the studied variables. Three to seven

factors and 7 or more random seeds were always applied in the PMF model. Thus, the

optimal number of factors/sources was 4 based on the robust and theoretical Q values

(Hegg et al., 2009, 2010). However, 3 factors provided more physically reasonable

results and the most easily identifiable sources, which was consistent with studies of

snow in northeastern China (Zhang et al., 2013a) and North America (Doherty et al.,

2014). The diagnostic regression $R^2$ value for $C_{BC}^{max}$ with this 3-factor solution was

considerably high (0.87). Rotational ambiguity was tested by varying the peak

parameter, which also indicated stable results. Hence, the 3-factor solution was the

best choice.

Figure 5 shows the source profile, including the measured mass concentrations (lines)

and the percent of species apportioned to each factor (dots) for the 3-factor solution.

The first factor (top panel) was obviously characterized by high loadings of $SO_4^{2-}$,

$NO_3^-$, $Cl^-$, and $NH_4^+$. $SO_4^{2-}$ and $NO_3^-$ are well-known markers for the burning of

fossil fuel, such as coal and oil (e.g., Xie et al., 2008; Oh et al., 2011; Zhang et al.,

2013b). $Cl^-$ is usually regarded as an important component of sea salt but also a

product of industrial emission (Kulkarni, 2009; Dall'Osto et al., 2013) and coal



combustion (Hailin et al., 2008). Furthermore, the molar ratio of $Cl^-$ to $Na^+$ was clearly larger than that of sea salt by a factor of $> 2$, which implied another source in addition to sea salt. Additionally, $NH_4^+$ is a recognized marker of coal combustion. However, the ratio of $C_{BC}^{max}$ to $SO_4^{2-}$ was low (0.04) and close to the values in

pollution sources as reported by Hegg et al. (2009). Therefore, we considered the first factor an industrial pollution source. The second factor (middle panel) presented substantial loadings of $Na^+$, $K^+$, and $K_{Biosmoke}$. However, chemical analyses are not available for certain organic matter, including levoglucosan, succinate, oxalate, and formate, which generally indicate biomass burning. $K^+$ and $K_{Biosmoke}$ are markers for

biomass burning emissions and were highly loaded for this factor. In particular, $K_{Biosmoke}$, which was calculated as the biosmoke fraction of K, was representative (Zhang et al., 2013a). Compared with the first factor, the molar ratio of $Cl^-$ to $Na^+$ was smaller than 1, which indicates that $Na^+$ was also a potential product of emission sources in addition to sea salt, such as biomass burning (Oh et al., 2011). However,

the ratio of $C_{BC}^{max}$ to $SO_4^{2-}$ was relatively high compared with that of the previously identified industrial pollution sources and close to the ratio for a biomass burning source as reported by Doherty et al. (2014). Therefore, we interpreted the second factor as a biomass burning source. The third factor (bottom panel) accounted for over 50% Al, Cr, Fe, Cu, and Ba in the samples. Al and Fe are well-known crustal

constituents, and they are usually used to determine the mass of soil dust. The mass ratio of Fe to Al (0.36) was close to the value in the continental crust (0.40) (Wedepohl, 1995). Furthermore, the enrichment factors (EFs) of the trace elements,

including Cr, Cu, and Ba, were < 5 at many sample sites, which indicates that these

elements may have originated from a crustal source (Zhang et al., 2013b). Hence, we

can interpret the third factor as a soil dust source.

### 3.4.2 Source contributions to the sample sites

Figure 6 shows the mean normalized contributions from each source to the individual

receptor site. The industrial pollution source with a 31% $C_{BC}^{max}$ dominated the

contributions to the sites in Region 3. In particular, the mean normalized contributions

were > 5 at sites 60 and 84, which was consistent with the abundant loadings of $SO_4^{2-}$

and $NO_3^-$. The results were not surprising because substantial industrial activity

occurs in this region. However, the contributions from the biomass burning source

with the highest $C_{BC}^{max}$ (53%) were more geographically dispersed and relatively

evenly distributed across Regions 1, 2, and 4. At sites 53 and 67, the contributions

were significantly large, which were likely caused by the concentrations of $K_{Biosmoke}$,

which were over 500 ng g$^{-1}$ and much higher than those at the other sites. Indeed,

biomass burning, such as biofuel combustion for heating, in winter and early spring in

northwestern China is normally prevalent (Pu et al., 2015). Unsurprisingly, a soil dust

source, which was characterized by the highest loadings of Al and Fe, was mainly

associated with the sites in Qinghai (Region 1), although the contributions were

obvious at certain sites in Xinjiang, especially the sites in Region 5, which was

partially because these sites were located on hills with scarce plants, and wind may

have blown local soil dust onto the snow.

### 3.4.3 Source attribution of the particulate absorption





The fractional contributions from the three sources to the 650-700-nm particulate absorption at each individual receptor site are shown in Figure 7. The average regional contributions are shown in Table 4. The most remarkable feature of the source attributions is the differences observed by region. Biomass burning was the

primary source in Region 1 (in Qinghai) and in Regions 2 and 4 (in Xinjiang), and it presented average regional contributions of 59%, 60%, and 67%, respectively. Although high dust mass was present in the snow samples from Region 1, this source attribution was reasonable because the $C_{BC}^{max}$ from the biomass burning sources was much larger than that from the soil dust sources by a factor of > 3 (Figure 6). In

particular, biomass burning in Qinghai is prevalent, especially during winter (Yan et al., 2006). In Region 1, soil dust accounted for 29% of the particulate absorption, which was less than the contribution from the biomass burning sources but more significant compared with the contributions from soil dust in the other regions. In Regions 2 and 4, most of the sample sites were located on mountains and far from

industrial areas; therefore, dominant absorption by biomass burning sources was not anomalous. The only exception was site 58, which was dominated by industrial pollution sources, and this result was likely because of its shorter distance from cities and lower elevation. In Region 3, all of the sample sites were located near cities and suffered from anthropogenic emissions; therefore, industrial pollution was the primary

source and presented a contribution of 58%. In Region 5, the primary source differed between sites. Absorption was dominated by biomass burning sources at sites 73 and 78, by industrial pollution sources at sites 72, 74, and 76, and by soil dust sources at

sites 75 and 77. More complex topography and emission sources could partly explain these findings.

The PMF results in Qinghai in this study were inconsistent with those by Zhang et al. (2013a), who indicated that soil dust was the dominant source of ILAPs. However, the

discrepancy was mild because (1) the receptor sites were far from each other, (2) the chemical species inputs were different, and (3) the variables that characterized the particulate light absorption in the PMF analysis were inconsistent. Hence, additional sample sites and more complete and advanced analyses of ILAPs and chemical species inputs to the PMF model are necessary to obtain more representative source

analysis results.

### 3.4.4 Altitude gradients of BC mass

Ye et al. (2012) performed a preliminary study on the same field campaign and found that the snow BC mixing ratios, which were based on visual estimates, were negatively correlated with the altitudes of the sample sites in Xinjiang. In our study,

the $C_{BC}^{est}$ and $C_{BC}^{max}$ values from the ISSW presented a similar trend (Figure 8a and b). Thus, altitude is an important influencing factor for BC mixing ratios and particulate absorption. Additionally, the $C_{BC}^{max}$ value simulated by the PMF model decreased steadily with altitude, although this trend was not as obvious as that for the measured $C_{BC}^{est}$ and $C_{BC}^{max}$. Therefore, we can explore the cause of altitude gradients for the BC

mass based on the PMF 3-factor solution. Figure 9 shows the contributions of each source as a function of altitude. The sample snow from site 53 was dirty (Ye et al., 2012); therefore, we did not consider the results of that site in the trend analysis.





Clearly, the contributions from industrial pollution sources presented a decreasing trend with altitude, whereas the contributions from biomass burning and dust soil sources did not show obvious gradient variations. Thus, the altitude gradients of the contributions from industrial pollution likely caused the altitude gradients of the BC mass, and the effects of biomass burning and soil dust were limited.

### 3.5 Mass contribution of the chemical components

In addition to performing a PMF analysis, the chemical components must be evaluated to examine the potential emission sources. As shown in Figure 10a, the median mass concentrations of chemicals in the different regions were 6.0 $\mu g\ g^{-1}$ (Region 1), 5.0 $\mu g\ g^{-1}$ (Region 2), 7.0 $\mu g\ g^{-1}$ (Region 3), 5.0 $\mu g\ g^{-1}$ (Region 4), and 4.6 $\mu g\ g^{-1}$ (Region 5). At sites 60 and 84, the mass concentrations were > 20 $\mu g\ g^{-1}$, which were primarily contributed by $SO_4^{2-}$ and $NO_3^-$. Regionally, the chemical components were dominated by MD (30.5%) and OC (29.5%) in Region 1, MD (21.1%) and $SO_4^{2-}$ (20.2%) in Region 2, $SO_4^{2-}$ (21.5%) and $NO_3^-$ (29.2%) in Region 3, $SO_4^{2-}$ (17.9%) and $NO_3^-$ (33.7%) in Region 4, and MD (29.5%) and $NO_3^-$ (27.5%) in Region 5 (Figure 10b). These results indicated that soil dust sources greatly contributed to the chemical components in Region 1, whereas industrial pollution was the predominant source in Regions 3 and 4. In the other regions, the fractional contributions from different sources to the mass concentrations were comparable. The mass contribution was not proportional to the absorption contribution because of the different ILAP loadings of emission sources. For example, although $SO_4^{2-}$ and $NO_3^-$ are primary species, the 650-700-nm particulate absorption in Region 4 was dominated by a



biomass burning source. BC is regarded as an important light-absorbing particle, and it ranged from 0.2 to 4.8% in mass at all sites, with an average of 1.3%, which is smaller than that (2.4-5.1%) in urban areas in China (Huang et al., 2014). $K_{Biosmoke}$ is a good indicator of biomass burning, and it ranged from 0.4 to 7.8% and presented the

largest regional average fractional contribution (3.0%) in Region 2. Wang et al. (2016) reported comparable values of 1.3-5.1% in the snow in northeastern China.

### 3.6 Comparative analysis of chemical components

Bond et al. (2004) indicated that the OC:BC ratio of the emissions from fossil fuel burning is lower than that from biomass or biofuel burning; therefore, we may

qualitatively examine the primary emission sources based on this theory. In this work, the regional average ratios of OC to BC were 20.9, 6.12, 3.99, 6.71, and 7.28 (Figure 11a). The smallest value in Region 3 was similar to those observed in Beijing (Zhang et al., 2013b). The similar ratios in Regions 2, 4, and 5 were close to that of savanna and grassland regions as reported by Andreae and Merlet (2001). The results

suggested the relative dominance of an industrial pollution source in Region 3 and a biomass burning source in other regions in Xinjiang. This pattern was similar to that of the source apportionment analysis by the PMF model. The largest value in Region 1 in Qinghai implied a primary contribution to OC from soil dust.

Sources of nitrate are considerably more varied than the sources of sulfate (Arimoto et

al., 1996). For $NO_3^-$, the largest source is fossil fuel combustion. Biomass burning is regarded as another main source, which was determined according to the analysis of Logan. Additionally, microbial activity in soil is a potential source of nitrate. However,



$SO_4^{2-}$ is mainly a product of burning coal. Thus, we can compare the correlation between $NO_3^-$ and $SO_4^{2-}$ to explore the variety of emission sources. As shown in Figure 11b, the average regional ratio of $NO_3^-$ to $SO_4^{2-}$ in Region 1 was 0.49, which was lower than that in the other regions. The concentrations and correlation coefficients of $NO_3^-$ and $SO_4^{2-}$ were both the lowest in Region 1, which indicates limited emissions from an industrial pollution source. In Xinjiang, the ratios mostly ranged from 1 to 1.5, and the high correlation coefficient in Region 5 was associated with similar industrial pollution sources, whereas the low correlation coefficients in Regions 2-4 may have been related to complicated industrial pollution sources.

A comparison between cations and anions is shown in Figure 11c. Generally, the correlation coefficients were all significant at the 5% level. A large anion charge deficit was observed in Region 1, which presented an average regional ratio of 2.93, which was likely caused by the absence of detected $CO_3^{2-}$ and $HCO_3^-$. Carbonates (e.g., $CaCO_3$ and $MgCO_3$) are often abundant in dust, which was observed in the Central Himalayan Glacier (Xu et al., 2013), and account for the largest contribution from soil dust to the mass concentration. However, the average ratios in the sample snow in Xinjiang were generally uniform and ranged from 0.7-1.2, which suggests an adequate charge balance. Overall, the concentrations of inorganic ions in the snow samples were lower than those in the rainwater in urban sites in China (Wang and Han, 2011) but larger than previous measurements in the Himalayas (Thompson et al., 2000; Xu et al., 2013).

**4 Discussion and conclusions**



A large field campaign was conducted in northwestern China from January to February 2012, with 284 snow samples collected from 38 sites in Xinjiang Province and 6 sites in Qinghai Province. Based on a previous study in North China (Wang et al., 2013), we estimated the ILAP content and analyzed the chemical components. In

addition, we used a PMF model to explore the fractional contributions from different sources to light absorption.

In Qinghai, the snow was thin and patchy; therefore, local soil could be lofted and deposited onto this snow. In this region, non-BC ILAPs predominantly contributed to light absorption. The cleanest snow of this campaign was found in the northeast of

Xinjiang along the border of China, and the lowest $C_{BC}^{est}$ values were approximately 5 ng g$^{-1}$. The highest $C_{BC}^{est}$ value was approximately 450 ng g$^{-1}$ at site 60, which was located in proximity to industrial cities. Although the sites that were located to the north of the Tianshan Mountains were more affected by human activities, the average $C_{BC}^{est}$ value was intermediate and comparable to the values in the other regions.

Overall, the $C_{BC}^{est}$ values of most of the snow samples ranged from 10 to 150 ng g$^{-1}$ in this campaign. In the Tianshan Mountains (Regions 2 and 3), the $Å_{tot}$ and $f_{nonBC}^{est}$ values presented vertical changes from the subsurface to surface snow, which indicates a probable shift in emission sources as winter progressed.

In Qinghai (Region 1), OC dominated the 450-nm absorption, and the fractional

contribution reached approximately 70%. Although dust accounted for the main portion of the particulate mass, the fractional contribution from Fe to absorption (approximately 5%) was not significant. In Xinjiang, the fractions of absorption from

BC (45%) and OC (50%) were comparable, and the effect of Fe was limited.

A 3-factor solution (industrial pollution, biomass burning and soil dust) was used to explore the fractional contributions from the different sources within the particulate absorption range from 650-700 nm based on a source apportionment analysis with the

PMF model. In Qinghai, biomass burning was the primary source (59%), and soil dust accounted for 29% of the particulate absorption despite its high mass contribution. In Xinjiang, the source attributions varied by region. In Regions 2 and 4, most of the sample sites were located on mountains and far from industrial areas, and biomass burning sources were dominant and accounted for 60% and 67% of the contributions,

respectively. In Region 3, absorption was dominated by an industrial pollution source (58%) because of the shorter distance from cities and lower elevations. In Region 5, the topography and emission sources were more complex, and the primary sources differed between sites. However, the BC mixing ratios showed a negative correlation with altitude in Xinjiang. An analysis based on the PMF 3-factor solution showed that

this observation likely resulted from gradient variations in the contributions from industrial pollution sources.

An evaluation indicated that the predominant source of chemical components was soil dust in Region 1 and industrial pollution in Regions 3 and 4. The mass of BC, which is regarded as an important light-absorbing particle, ranged from 0.2 to 4.8% and

presented an average of 1.3%. A comparison between OC and BC suggested the relative dominance of an industrial pollution source in Region 3 and a biomass burning source in other regions in Xinjiang. The relationship between $NO_3^-$ and $SO_4^{2-}$



also showed limited emissions from an industrial pollution source in Qinghai but

complicated sources in Xinjiang. Finally, a large anion charge deficit was observed in

Region 1, which explained the predominant contribution from soil dust to the mass

concentration.





*Acknowledgements.* This research was supported by the Foundation for Innovative Research Groups of the National Science Foundation of China (41521004), the National Science Foundation of China under Grants 41522505, and the Fundamental Research Funds for the Central Universities (lzujbky-2015-k01, lzujbky-2016-k06 and lzujbky-2015-3). We thank Jinsen Shi, Hao Ye of Lanzhou University and Rudong Zhang of Nanjing University for their assistance in field sampling.





**Table 1.** Statistics of the seasonal snow variables measured using an ISSW for each site.

| Site | Layer | Latitude (N) | Longitude (E) | Average snow depth (cm) | Sample depth (cm) Top | Sample depth (cm) Bottom | $C_{BC}^{equiv}$ (ng g$^{-1}$) | $C_{BC}^{max}$ (ng g$^{-1}$) | $C_{BC}^{est}$ (ng g$^{-1}$) | $f_{non-BC}^{est}$ (%) | $Å_{tot}$ |
|---|---|---|---|---|---|---|---|---|---|---|---|
| 47 | 1 | 35.54 | 99.49 | 2.5 | 0 | 5 | 480 | 242 | 148(—,235) | 73(55,—) | 4.58 |
| | 2 | | | | 5 | 10 | 487 | 266 | 175(—,262) | 64(46,—) | 4.02 |
| | 3 | | | | 10 | 15 | 1094 | 548 | 358(—,556) | 67(49,—) | 4.30 |
| | 4 | | | | 15 | 20 | 314 | 200 | 156(29,217) | 52(33,—) | 3.55 |
| 48 | 1 | 34.85 | 98.13 | 4.5 | 0 | 1 | 334 | 196 | 152(65,222) | 58(37,—) | 3.87 |
| | 2 | | | | 2 | 4 | 237 | 129 | 83(—,134) | 65(43,—) | 4.09 |
| 49 | 1 | 35.22 | 98.95 | 10 | 0 | 1 | 317 | 160 | 118(—,179) | 66(47,—) | 4.36 |
| | 2 | | | | 2 | 7 | — | — | —(—,—) | —(—,—) | — |
| | 3 | | | | 11 | 16 | 315 | 156 | 102(—,164) | 64(42,—) | 4.38 |
| | 4 | | | | 16 | 25 | 203 | 90 | 53(—,84) | 74(59,—) | 4.20 |
| 50 | 1 | 34.80 | 99.05 | — | 0 | 2 | 165 | 66 | 28(—,54) | 83(67,—) | 4.57 |
| | 2 | | | | 2 | 7 | — | — | —(—,—) | —(—,—) | — |
| | 3 | | | | 7 | 12 | — | — | —(—,—) | —(—,—) | — |
| | 4 | | | | 12 | 20 | 202 | 105 | 63(—,107) | 69(47,—) | 4.37 |
| 51a | 1 | 33.89 | 99.80 | 5 | 0 | 4 | 318 | 183 | 135(19,193) | 55(36,—) | 3.91 |
| | 2 | | | | 4 | 6 | — | — | —(—,—) | —(—,—) | — |
| 51b | 1 | 33.89 | 99.80 | 5 | 0 | 1 | 389 | 148 | 77(—,125) | 80(68,—) | 4.27 |
| 52 | 1 | 34.92 | 100.89 | 3.5 | 0 | 1 | 144 | 65 | 33(—,56) | 77(61,—) | 4.09 |
| | 2 | | | | 2 | 4 | — | — | —(—,—) | —(—,—) | — |
| 53 | 1 | 43.07 | 86.81 | 7 | 0 | 8 | 595 | 427 | 334(201,48 | 44(19,66) | 2.50 |
| | 2 | | | | 8 | 16 | 489 | 354 | 264(173,40 | 46(18,65) | 2.46 |
| | 3 | | | | 16 | 26 | 522 | 376 | 254(177,42 | 51(20,67) | 2.51 |
| 54 | 1 | 43.08 | 85.82 | 25 | 0 | 2 | 36 | 18 | 7(—,13) | 81(65,—) | 4.48 |
| | 2 | | | | 2 | 9 | 25 | 15 | 6(—,10) | 75(58,—) | 4.15 |



| Site | Layer | Latitude (N) | Longitude (E) | Average snow depth (cm) | Sample depth (cm) Top | Bottom | $C_{BC}^{equiv}$ (ng g⁻¹) | $C_{BC}^{max}$ (ng g⁻¹) | $C_{BC}^{est}$ (ng g⁻¹) | $f_{non-BC}^{est}$ (%) | $\text{Å}_{tot}$ |
|---|---|---|---|---|---|---|---|---|---|---|---|
| | 3 | | | | 12 | 17 | 111 | 86 | 68(25,87) | 39(22,78) | 2.91 |
| 55 | 1 | 43.51 | 83.54 | 25 | 0 | 4 | 146 | 110 | 96(52,122) | 32(30,63) | 2.57 |
| | 2 | | | | 4 | 8 | 111 | 111 | 97(61,120) | 13(7,45) | 2.22 |
| 56 | 1 | 43.66 | 82.75 | 24 | 0 | 4 | 90 | 61 | 45(16,59) | 50(34,82) | 2.92 |
| | 2 | | | | 4 | 8 | 162 | 114 | 99(47,126) | 43(27,74) | 2.72 |
| | 3 | | | | 8 | 12 | — | — | —(—,—) | —(—,—) | — |
| 57 | 1 | 43.64 | 82.11 | 15 | 0 | 4 | 151 | 124 | 106(57,133) | 40(25,58) | 2.52 |
| | 2 | | | | 4 | 9 | 106 | 78 | 62(34,77) | 42(27,68) | 2.42 |
| | 3 | | | | 9 | 13 | — | — | —(—,—) | —(—,—) | — |
| 58 | 1 | 43.52 | 81.13 | 37 | 0 | 4 | 37 | 26 | 10(1,21) | 73(43,97) | 3.38 |
| | 2 | | | | 4 | 8 | 50 | 35 | 18(8,30) | 64(38,83) | 2.91 |
| | 3 | | | | 8 | 13 | 118 | 83 | 63(42,85) | 47(29,64) | 2.25 |
| | 4 | | | | 13 | 18 | 61 | 32 | 15(5,28) | 79(53,—) | 3.51 |
| | 5 | | | | 18 | 23 | 70 | 42 | 25(13,39) | 66(43,84) | 2.82 |
| | 6 | | | | 25 | 30 | 141 | 133 | 117(85,147) | 17(5,40) | 1.96 |
| 59 | 1 | 44.49 | 81.15 | 60 | 0 | 5 | 116 | 51 | 24(—,38) | 80(68,—) | 4.29 |
| | 2 | | | | 5 | 10 | 109 | 37 | 16(—,27) | 85(75,—) | 4.41 |
| | 3 | | | | 10 | 15 | 195 | 99 | 67(15,91) | 65(53,98) | 3.45 |
| | 4 | | | | 15 | 20 | 175 | 98 | 73(23,95) | 58(46,87) | 2.99 |
| | 5 | | | | 20 | 28 | 184 | 97 | 71(19,93) | 62(50,90) | 3.08 |
| | 6 | | | | 28 | 36 | 204 | 102 | 67(—,93) | 67(54,—) | 3.60 |
| 60 | 1 | 44.96 | 82.63 | 5 | 0 | 2 | 696 | 542 | 473(352,58) | 32(17,49) | 1.84 |
| | 2 | | | | 2 | 5 | — | — | —(—,—) | —(—,—) | — |
| 61 | 1 | 44.38 | 83.09 | 23 | 0 | 3 | 192 | 113 | 83(37,107) | 57(45,81) | 2.73 |
| | 2 | | | | 3 | 6 | 286 | 197 | 156(56,200) | 45(30,80) | 2.88 |





| Site | Layer | Latitude (N) | Longitude (E) | Average snow depth (cm) | Sample depth (cm) Top | Sample depth (cm) Bottom | $C_{BC}^{equiv}$ (ng g⁻¹) | $C_{BC}^{max}$ (ng g⁻¹) | $C_{BC}^{est}$ (ng g⁻¹) | $f_{non\text{-}BC}^{est}$ (%) | $\text{Å}_{tot}$ |
|---|---|---|---|---|---|---|---|---|---|---|---|
| | 3 | | | | 6 | 15 | 260 | 144 | 110(47,141) | 58(46,82) | 2.74 |
| | 4 | | | | 15 | 19 | 94 | 69 | 45(18,60) | 52(36,81) | 2.85 |
| 62 | 1 | 44.57 | 83.96 | 8 | 0 | 1 | 105 | 71 | 50(26,64) | 53(38,77) | 2.58 |
| | 2 | | | | 1 | 4 | 144 | 80 | 56(24,74) | 61(49,84) | 2.79 |
| 63 | 1 | 45.58 | 84.29 | 10 | 0 | 1 | 169 | 109 | 82(40,108) | 52(37,77) | 2.69 |
| 64 | 1 | 46.68 | 83.54 | 25 | 0 | 2 | 325 | 199 | 152(38,205) | 51(34,94) | 3.41 |
| | 2 | | | | 2 | 6 | 269 | 120 | 84(11,114) | 69(58,96) | 3.31 |
| | 3 | | | | 6 | 10 | 893 | 363 | 240(—,336) | 73(63,—) | 3.75 |
| | 4 | | | | 10 | 15 | 268 | 125 | 75(—,106) | 72(61,—) | 3.61 |
| 65 | 1 | 46.49 | 85.04 | 4 | 0 | 2 | 156 | 108 | 83(33,106) | 46(31,79) | 2.85 |
| 66 | 1 | 46.88 | 85.92 | 15 | 0 | 5 | 57 | 40 | 24(1,33) | 58(42,—) | 3.48 |
| | 2 | | | | 5 | 10 | 134 | 98 | 75(22,96) | 46(30,85) | 3.06 |
| 67 | 1 | 47.26 | 86.71 | 7 | 0 | 4 | 611 | 446 | 386(379,48) | 38(22,72) | 2.73 |
| | 2 | | | | 4 | 10 | 298 | 234 | 192(96,243) | 38(21,71) | 2.68 |
| | 3 | | | | 10 | 16 | 256 | 195 | 140(48,183) | 45(28,81) | 2.93 |
| 68 | 1 | 48.15 | 86.56 | 32 | 0 | 5 | 140 | 120 | 87(46,112) | 38(20,67) | 2.52 |
| | 2 | | | | 5 | 10 | 84 | 70 | 46(21,62) | 45(26,75) | 2.72 |
| | 3 | | | | 10 | 15 | 104 | 82 | 58(26,77) | 44(26,75) | 2.74 |
| | 4 | | | | 15 | 20 | 20 | 15 | 6(1,12) | 70(40,95) | 3.33 |
| 69 | 1 | 47.86 | 86.29 | 4 | 0 | 1 | 320 | 264 | 208(137,26) | 36(19,59) | 2.22 |
| | 2 | | | | 1 | 4 | 191 | 155 | 112(64,147) | 42(23,67) | 2.45 |
| 70 | 1 | 48.33 | 87.13 | 70 | 0 | 3 | 39 | 24 | 12(—,18) | 68(53,—) | 3.93 |
| | 2 | | | | 3 | 8 | 31 | 21 | 13(0,17) | 59(44,99) | 3.46 |
| | 3 | | | | 10 | 15 | 42 | 29 | 18(2,24) | 57(42,96) | 3.37 |
| | 4 | | | | 15 | 20 | 9 | 6 | 3(—,4) | 68(51,—) | 3.69 |





| Site | Layer | Latitude (N) | Longitude (E) | Average snow depth (cm) | Sample depth (cm) Top | Bottom | $C_{BC}^{equiv}$ (ng g⁻¹) | $C_{BC}^{max}$ (ng g⁻¹) | $C_{BC}^{est}$ (ng g⁻¹) | $f_{non-BC}^{est}$ (%) | $\text{Å}_{tot}$ |
|---|---|---|---|---|---|---|---|---|---|---|---|
| | 5 | | | | 20 | 25 | 18 | 13 | 8(1,10) | 57(41,94) | 3.30 |
| | 6 | | | | 25 | 30 | 29 | 25 | 16(7,21) | 43(27,75) | 2.74 |
| | 7 | | | | 30 | 35 | 21 | 17 | 11(3,14) | 50(34,85) | 3.03 |
| | 8 | | | | 35 | 40 | 36 | 28 | 19(6,24) | 48(32,83) | 2.97 |
| 71 | 1 | 48.07 | 87.03 | 40 | 0 | 5 | 96 | 76 | 52(21,69) | 45(29,78) | 2.81 |
| | 2 | | | | 5 | 10 | 88 | 70 | 48(20,63) | 45(28,78) | 2.81 |
| | 3 | | | | 13 | 18 | 30 | 23 | 13(2,19) | 58(39,92) | 3.26 |
| | 4 | | | | 20 | 25 | 33 | 27 | 17(7,23) | 50(30,80) | 2.87 |
| | 5 | | | | 25 | 30 | 128 | 101 | 85(60,104) | 33(18,53) | 1.96 |
| 72 | 1 | 47.79 | 87.56 | 20 | 0 | 3 | 272 | 213 | 135(45,185) | 50(32,83) | 2.97 |
| | 2 | | | | 3 | 8 | 167 | 135 | 90(47,125) | 47(26,72) | 2.64 |
| | 3 | | | | 8 | 12 | 88 | 71 | 40(18,62) | 56(30,80) | 2.87 |
| 73 | 1 | 47.55 | 88.61 | 41 | 0 | 4 | 101 | 73 | 50(15,66) | 51(35,86) | 3.04 |
| | 2 | | | | 4 | 9 | 108 | 79 | 56(21,73) | 49(33,81) | 2.88 |
| | 3 | | | | 10 | 15 | 42 | 30 | 17(2,24) | 61(43,98) | 3.43 |
| | 4 | | | | 15 | 20 | 45 | 34 | 21(5,28) | 55(37,90) | 3.16 |
| | 5 | | | | 20 | 26 | 91 | 73 | 52(24,66) | 43(27,74) | 2.69 |
| | 6 | | | | 26 | 32 | 77 | 63 | 44(21,57) | 43(26,73) | 2.67 |
| 74 | 1 | 47.63 | 88.40 | 34 | 0 | 4 | 174 | 139 | 115(73,142) | 34(30,59) | 2.24 |
| | 2 | | | | 4 | 9 | 129 | 95 | 70(35,90) | 46(31,73) | 2.59 |
| | 3 | | | | 10 | 15 | 76 | 61 | 41(17,54) | 46(28,77) | 2.78 |
| | 4 | | | | 15 | 21 | 73 | 58 | 39(16,51) | 47(30,78) | 2.82 |
| | 5 | | | | 21 | 25 | 83 | 67 | 46(21,60) | 44(27,75) | 2.70 |
| 75 | 1 | 47.58 | 88.78 | 40 | 0 | 5 | 45 | 29 | 10(—,23) | 80(50,—) | 3.81 |
| | 2 | | | | 5 | 10 | 38 | 25 | 7(—,19) | 82(50,—) | 3.74 |



| Site | Layer | Latitude (N) | Longitude (E) | Average snow depth (cm) | Sample depth (cm) Top | Sample depth (cm) Bottom | $C_{BC}^{equiv}$ (ng g$^{-1}$) | $C_{BC}^{max}$ (ng g$^{-1}$) | $C_{BC}^{est}$ (ng g$^{-1}$) | $f_{non-BC}^{est}$ (%) | $Å_{tot}$ |
|---|---|---|---|---|---|---|---|---|---|---|---|
| | 3 | | | | 10 | 15 | — | — | —(—,—) | —(—,—) | — |
| | 4 | | | | 15 | 22 | 22 | 16 | 4(1,13) | 84(40,94) | 3.29 |
| | 5 | | | | 24 | 29 | 78 | 53 | 33(18,49) | 58(36,77) | 2.66 |
| | 6 | | | | 30 | 35 | 45 | 35 | 18(8,30) | 61(32,81) | 2.90 |
| | 7 | | | | 35 | 40 | 29 | 21 | 7(2,17) | 77(39,93) | 3.28 |
| | 8 | | | | 40 | 45 | 27 | 19 | 5(1,15) | 81(42,97) | 3.40 |
| 76 | 1 | 47.17 | 88.70 | 26 | 0 | 2 | 315 | 218 | 144(77,202) | 54(35,76) | 2.63 |
| | 2 | | | | 2 | 6 | 229 | 168 | 113(78,159) | 51(30,67) | 2.30 |
| | 3 | | | | 6 | 11 | 210 | 173 | 110(63,158) | 48(25,70) | 2.56 |
| | 4 | | | | 11 | 16 | 93 | 74 | 32(15,62) | 66(34,84) | 2.99 |
| 77 | 1 | 47.27 | 89.97 | 45 | 0 | 4 | 53 | 39 | 17(5,33) | 68(39,91) | 3.21 |
| | 2 | | | | 5 | 10 | 40 | 29 | 11(3,24) | 74(40,92) | 3.23 |
| | 3 | | | | 10 | 15 | 20 | 14 | 1(0,11) | 97(44,99) | 3.46 |
| | 4 | | | | 15 | 20 | — | — | —(—,—) | —(—,—) | — |
| | 5 | | | | 20 | 25 | 21 | 16 | 2(1,12) | 91(41,94) | 3.29 |
| | 6 | | | | 25 | 30 | 19 | 14 | 1(1,11) | 96(44,98) | 3.44 |
| | 7 | | | | 30 | 35 | 17 | 14 | 1(1,11) | 96(40,92) | 3.22 |
| | 8 | | | | 35 | 40 | — | — | —(—,—) | —(—,—) | — |
| | 9 | | | | 40 | 45 | — | — | —(—,—) | —(—,—) | — |
| 78 | 1 | 46.85 | 90.32 | 35 | 0 | 2 | 147 | 75 | 44(3,63) | 71(57,—) | 3.73 |
| | 2 | | | | 2 | 7 | 39 | 24 | 10(—,17) | 75(56,—) | 4.01 |
| | 3 | | | | 7 | 12 | 23 | 16 | 7(—,12) | 71(47,—) | 3.60 |
| | 4 | | | | 12 | 17 | 33 | 22 | 11(—,18) | 67(46,—) | 3.55 |
| | 5 | | | | 17 | 22 | 13 | 10 | 4(—,7) | 76(46,—) | 3.55 |
| | 6 | | | | 22 | 27 | 30 | 21 | 11(0,17) | 65(44,98) | 3.45 |



| Site | Layer | Latitude (N) | Longitude (E) | Average snow depth (cm) | Sample depth (cm) Top | Bottom | $C_{BC}^{equiv}$ (ng g$^{-1}$) | $C_{BC}^{max}$ (ng g$^{-1}$) | $C_{BC}^{est}$ (ng g$^{-1}$) | $f_{non-BC}^{est}$ (%) | $Å_{tot}$ |
|---|---|---|---|---|---|---|---|---|---|---|---|
| 79 | 1 | 43.53 | 89.74 | 30 | 0 | 6 | 78 | 41 | 21(—,33) | 74(58,—) | 4.04 |
|  | 2 |  |  |  | 6 | 11 | 62 | 38 | 20(—,30) | 67(50,—) | 3.75 |
|  | 3 |  |  |  | 11 | 18 | 68 | 41 | 21(—,32) | 69(53,—) | 3.89 |
|  | 4 |  |  |  | 18 | 24 | 84 | 50 | 28(—,40) | 67(52,—) | 3.71 |
|  | 5 |  |  |  | 24 | 30 | 83 | 54 | 32(4,45) | 61(45,95) | 3.33 |
|  | 6 |  |  |  | 30 | 35 | 171 | 116 | 86(42,110) | 49(35,75) | 2.65 |
|  | 7 |  |  |  | 35 | 40 | 165 | 99 | 66(21,87) | 60(47,87) | 2.98 |
| 80 | 1 | 44.10 | 87.49 | 18 | 0 | 3 | 176 | 109 | 67(29,95) | 63(47,94) | 3.28 |
|  | 2 |  |  |  | 3 | 7 | 50 | 33 | 12(—,25) | 76(51,—) | 3.75 |
| 81 | 1 | 43.60 | 87.51 | 6 | 0 | 3 | 172 | 89 | 51(8,74) | 71(57,—) | 3.66 |
|  | 2 |  |  |  | 3 | 6 | 189 | 89 | 53(1,76) | 72(60,—) | 3.50 |
| 82 | 1 | 44.09 | 84.80 | 13 | 0 | 5 | 48 | 24 | 9(—,15) | 81(68,—) | 4.66 |
|  | 2 |  |  |  | 5 | 9 | 28 | 16 | 8(—,11) | 73(59,—) | 4.22 |
| 83 | 1 | 43.93 | 85.41 | 12 | 0 | 3 | 101 | 66 | 37(3,53) | 64(48,—) | 3.54 |
|  | 2 |  |  |  | 3 | 7 | 48 | 31 | 14(—,23) | 71(52,—) | 3.81 |
|  | 3 |  |  |  | 8 | 12 | 1087 | 841 | 619(336,76) | 43(29,69) | 2.44 |
| 84 | 1 | 43.93 | 86.76 | 25 | 0 | 2 | 489 | 372 | 250(127,34) | 49(30,74) | 2.63 |
|  | 2 |  |  |  | 2 | 6 | 76 | 50 | 7(—,38) | 91(49,—) | 3.68 |





**Table 2.** Surface and subsurface snow sample average values within the five sample regions in Figure 1.

| Regions | Layers | $C_{BC}^{max}$ | $C_{BC}^{est}$ | $C_{BC}^{equiv}$ | $Å_{tot}$ | $f_{non-BC}^{est}$ |
|---------|--------|--------|--------|--------|-------|--------|
|  |  | (ng g$^{-1}$) | (ng g$^{-1}$) | (ng g$^{-1}$) |  | (%) |
| 1 | surface | 151 ±66 | 99 ±53 | 307±119 | 4.24 ±0.29 | 70 ±11 |
|  | subsurface | 174 ±110 | 113 ±78 | 332 ±201 | 4.17 ±0.19 | 66 ±4 |
| 2 | surface | 108 ±126 | 81 ±102 | 160±171 | 3.27 ±0.80 | 59 ±19 |
|  | subsurface | 119 ±96 | 89 ±69 | 170±134 | 2.86 ±0.49 | 49 ±16 |
| 3 | surface | 173 ±183 | 127 ±158 | 245 ±226 | 3.11 ±0.86 | 58 ±15 |
|  | subsurface | 117 ±158 | 75 ±120 | 175 ±201 | 3.51 ±0.50 | 72 ±12 |
| 4 | surface | 160 ±141 | 126 ±124 | 218 ±192 | 2.99 ±0.56 | 48 ±11 |
|  | subsurface | 114 ±77 | 82 ±56 | 178 ±157 | 2.96 ±0.36 | 51 ±10 |
| 5 | surface | 112 ±79 | 74 ±56 | 158±104 | 3.09 ±0.56 | 58 ±16 |
|  | subsurface | 61 ±46 | 37 ±31 | 80 ±57 | 3.03 ±0.37 | 62 ±16 |



**Table 3.** Total snow water equivalent (SWE) and estimated total snowpack BC mass in a 1-cm$^2$ column of snow.

| Site | Date sampled | Integrated SWE | Integrated BC | Snowpack average BC mixing ratio |
|------|------|------|------|------|
| | (2012) | (g cm$^{-2}$) | (ng cm$^{-2}$) | (ng g$^{-1}$) |
| 47 | 10 Jan | 5.94 | 1212 | 204 |
| 48 | 11 Jan | 0.51 | 54 | 106 |
| 49 | 11 Jan | 3.77 | — | 74 |
| 50 | 1 Jan | 1.56 | — | 57 |
| 51a | 12 Jan | 1.52 | — | 135 |
| 51b | 12 Jan | 0.14 | 11 | 77 |
| 52 | 13 Jan | 0.14 | — | 33 |
| 53 | 31 Jan | 9.18 | 2632 | 287 |
| 54 | 1 Feb | 2.41 | 64 | 27 |
| 55 | 2 Feb | 1.16 | 112 | 96 |
| 56 | 2 Feb | 1.16 | — | 76 |
| 57 | 2 Feb | 1.99 | — | 80 |
| 58 | 3 Feb | 5.33 | 244 | 46 |
| 59 | 4 Feb | 7.67 | 451 | 59 |
| 60 | 4 Feb | 0.22 | — | 473 |
| 61 | 5 Feb | 2.87 | 265 | 92 |
| 62 | 8 Feb | 0.56 | 30 | 54 |
| 63 | 9 Feb | — | — | — |
| 64 | 9 Feb | 2.2 | 283 | 129 |
| 65 | 10 Feb | — | — | — |
| 66 | 10 Feb | 1.45 | 80 | 55 |
| 67 | 11 Feb | 5.64 | 1320 | 234 |
| 68 | 11 Feb | 2.9 | 128 | 44 |
| 69 | 11 Feb | — | — | — |
| 70 | 12 Feb | 7.9 | 99 | 13 |
| 71 | 12 Feb | 5.15 | 226 | 44 |
| 72 | 14 Feb | 1.74 | 138 | 80 |
| 73 | 14 Feb | 5.96 | 236 | 40 |
| 74 | 14 Feb | 4.16 | 235 | 56 |
| 75 | 17 Feb | 7.64 | — | 12 |
| 76 | 18 Feb | 2.68 | 230 | 86 |
| 77 | 19 Feb | 6.1 | — | 4 |
| 78 | 20 Feb | 5.07 | 53 | 10 |
| 79 | 21 Feb | 7.05 | 282 | 40 |
| 80 | 23 Feb | 0.91 | 32 | 35 |
| 81 | 23 Feb | 1.02 | 53 | 52 |
| 82 | 24 Feb | 0.97 | 8 | 8 |
| 83 | 24 Feb | 1.43 | 343 | 240 |
| 84 | 25 Feb | 0.66 | 58 | 88 |



**Table 4.** Average regional contributions to the 650-700-nm particulate absorption.

| Regions | Industrial pollution (%) | Biomass burning (%) | Soil dust (%) |
|---------|--------------------------|---------------------|---------------|
| 1 | 12 | 59 | 29 |
| 2 | 25 | 60 | 15 |
| 3 | 58 | 31 | 10 |
| 4 | 27 | 67 | 5 |
| 5 | 41 | 28 | 31 |





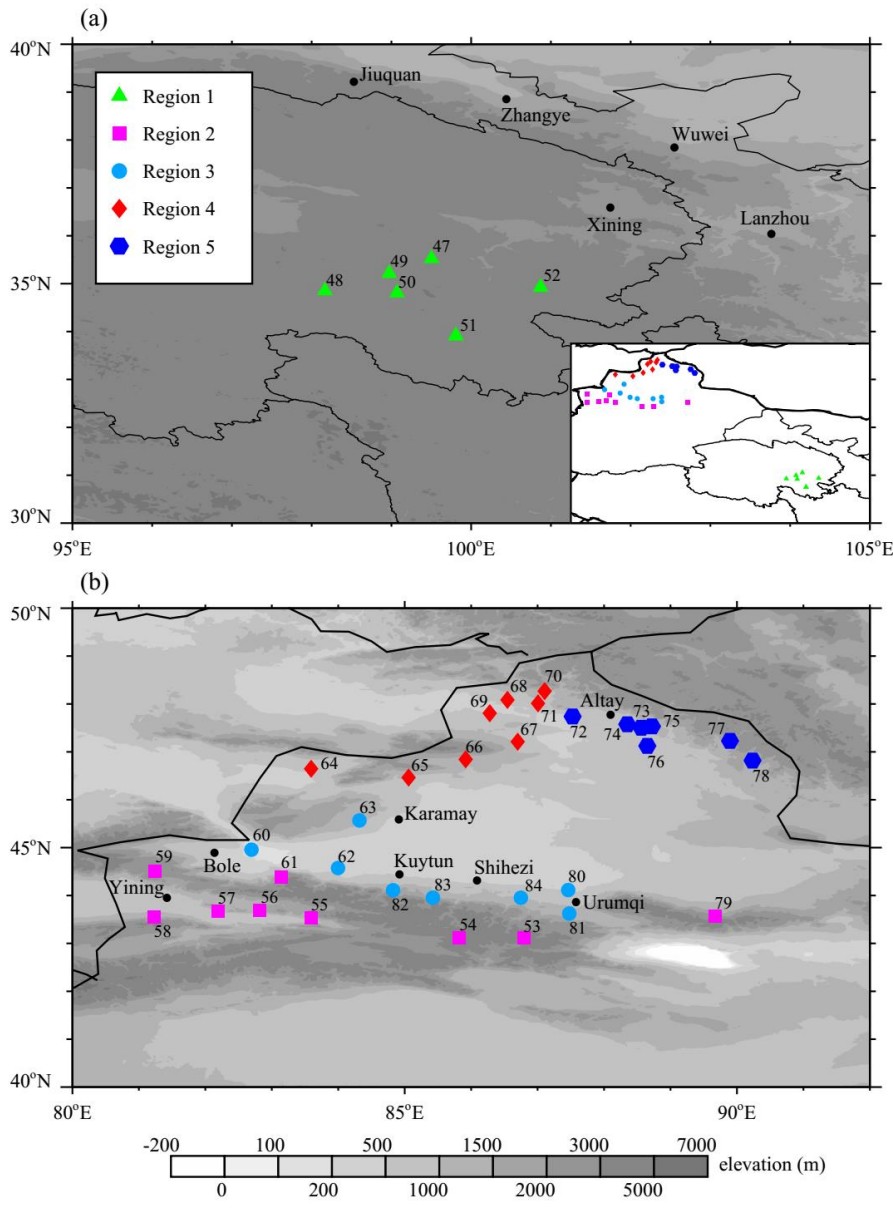

**Fig. 1.** Snow sampling locations, site numbers, and regional groupings in (a) Qinghai and (b) Xinjiang, black filled circle are the locations of main cities on the map.





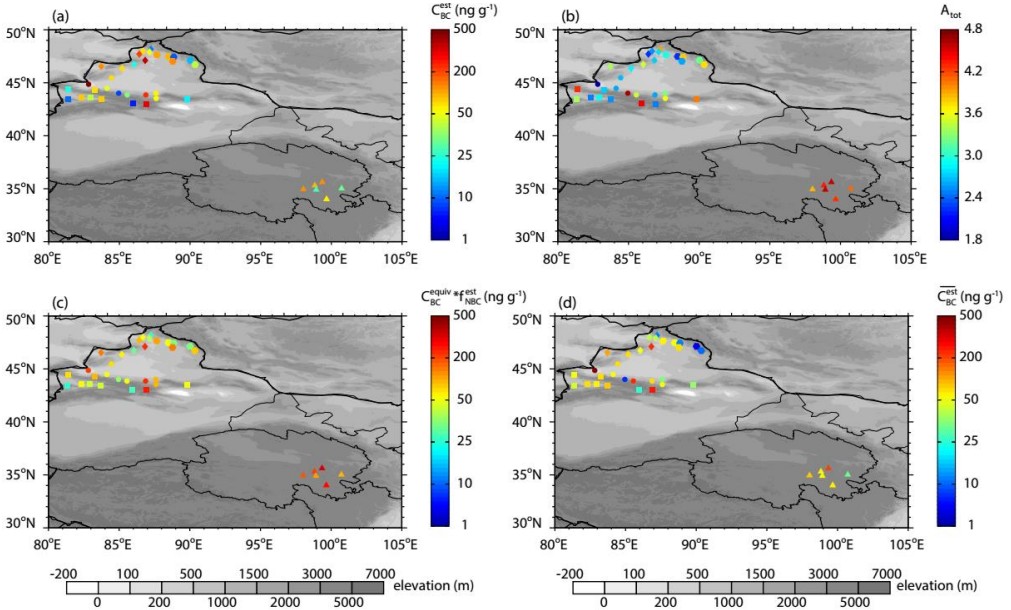

**Fig. 2.** (a) $C_{BC}^{est}$ and (b) $Å_{tot}$ for the surface layer at each site, with symbols according to region (see Figure 1). (c) Estimated BC-equivalent mixing ratio (ng g⁻¹) required to explain the spectrally integrated (300–750 nm) absorption of sunlight by non-BC components in snow. (d) Estimated average snow BC mixing ratio, $\overline{C_{BC}^{est}}$, which was calculated by integrating the snow water content and BC mass over the entire snowpack (see Table 3).





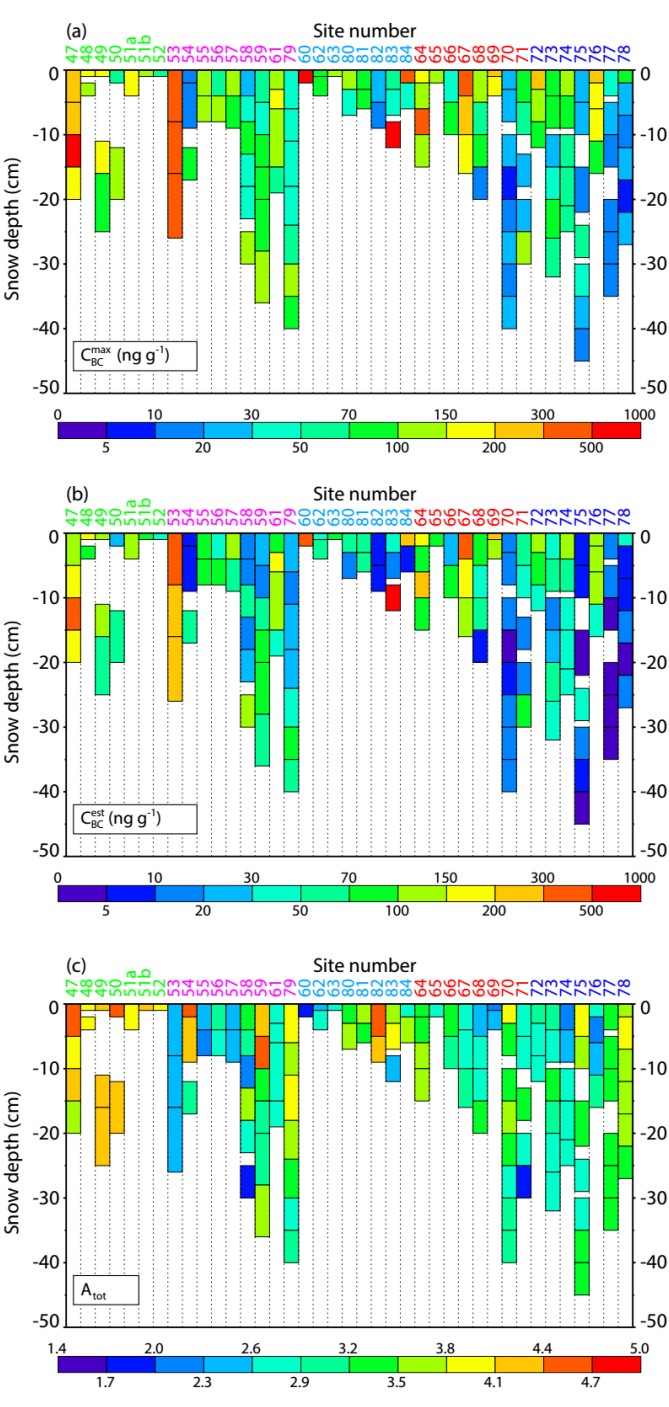

**Fig. 3.** Vertical profiles of the (a) $C_{BC}^{max}$, (b) $C_{BC}^{est}$, and (c) $Å_{tot}$ at each sampling site.



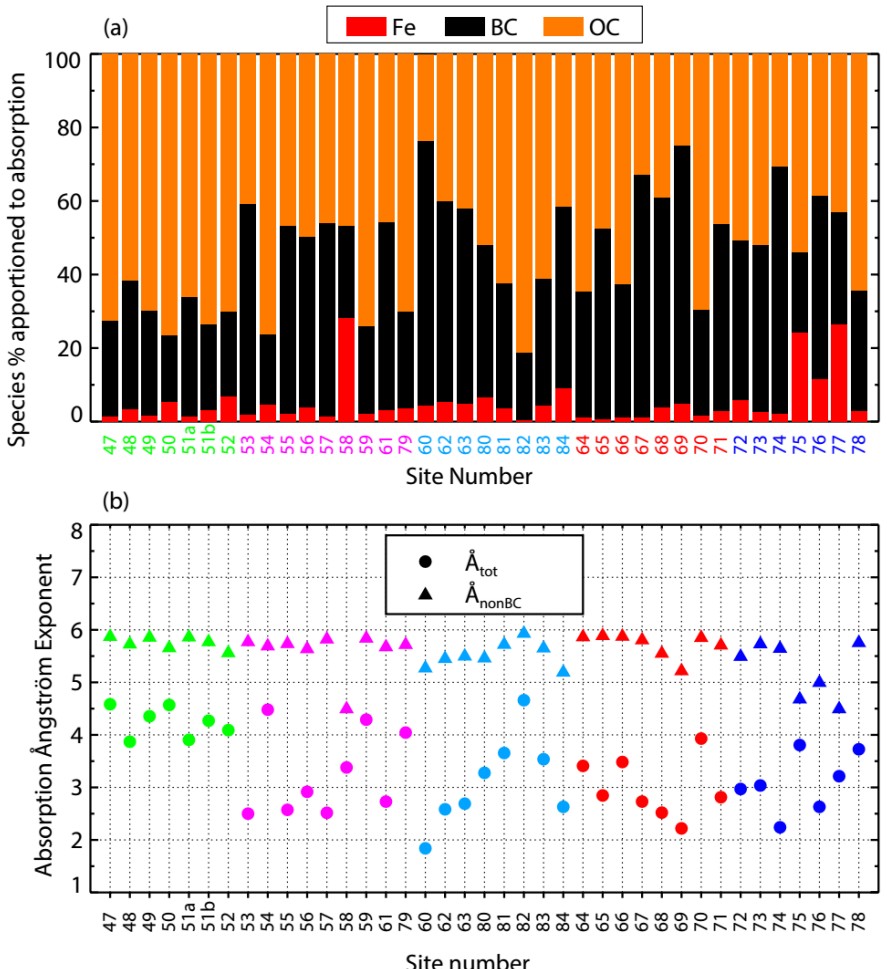

**Fig. 4.** (a) Relative contributions from the BC, OC, and Fe oxides to the total absorption optical depth for the surface snow samples. (b) $\text{Å}_{tot}$ and $\text{Å}_{nonBC}$ for the surface snow samples.





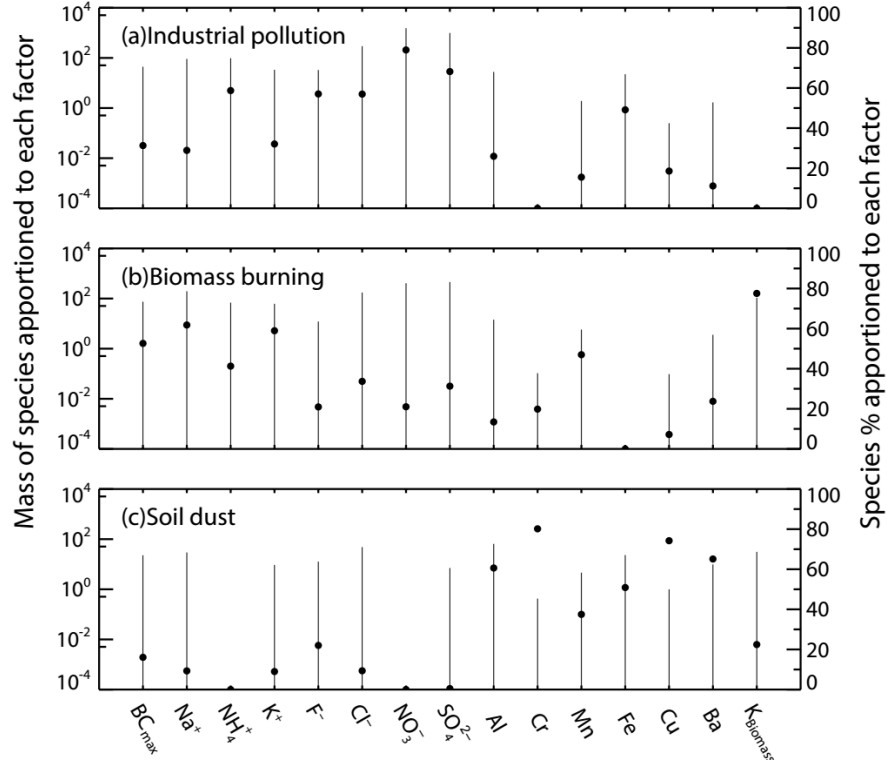

**Fig. 5.** Source profiles including the measured mass concentrations (lines) and the percent of species (dots) for the three factors/sources that were resolved by the PMF 5.0 model.





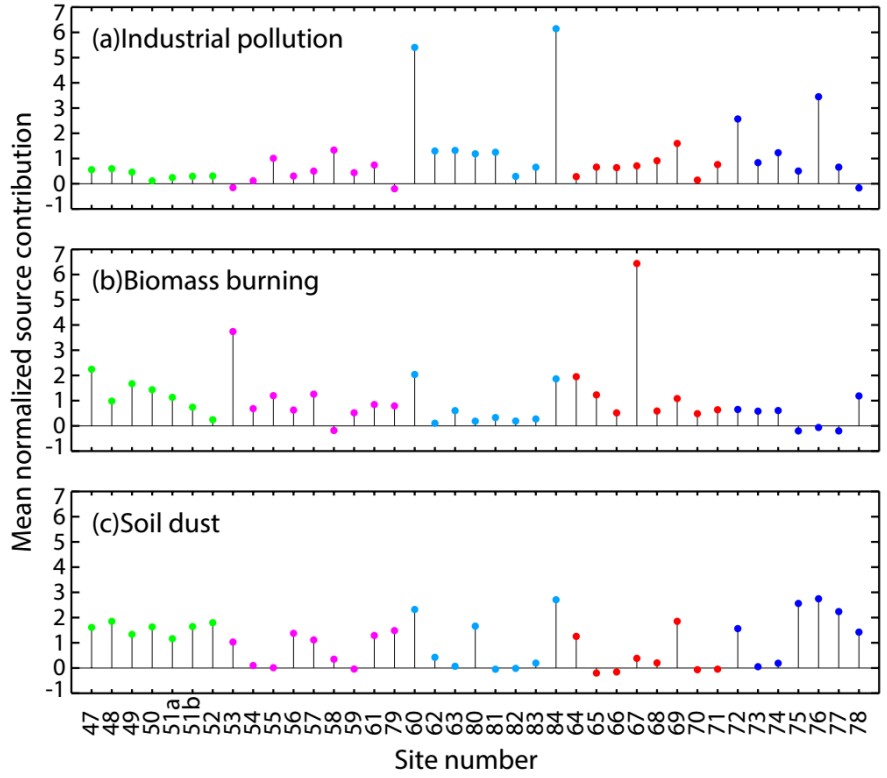

**Fig. 6.** Contributions from each source/factor to each sample or individual receptor site. The contributions have been normalized by the average value of the respective factor contributions over all sites.





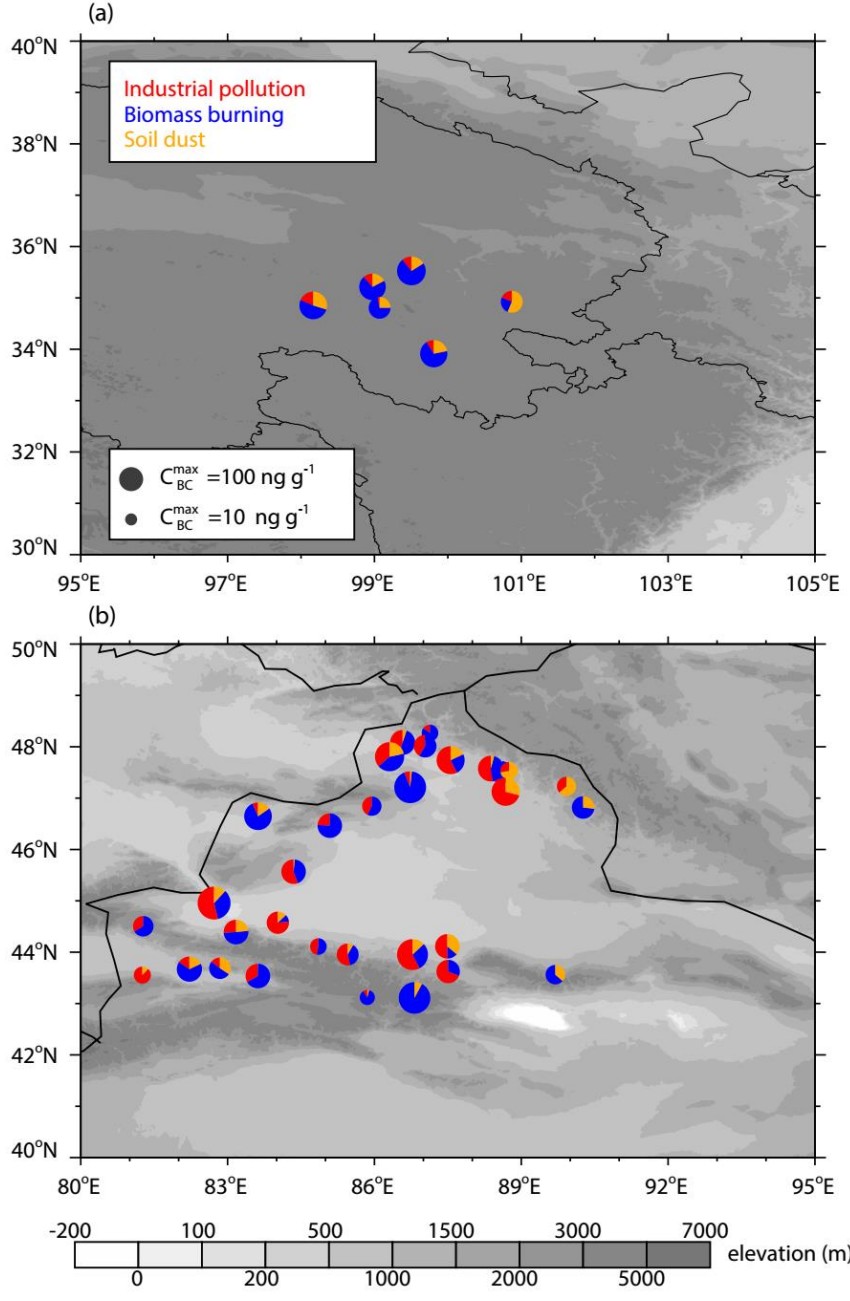

**Fig. 7.** Fractional contributions from soil dust, biomass burning, and industrial pollution to 650–700-nm particulate absorption according to the PMF analysis for the surface snow samples in (a) Qinghai and (b) Xinjiang.





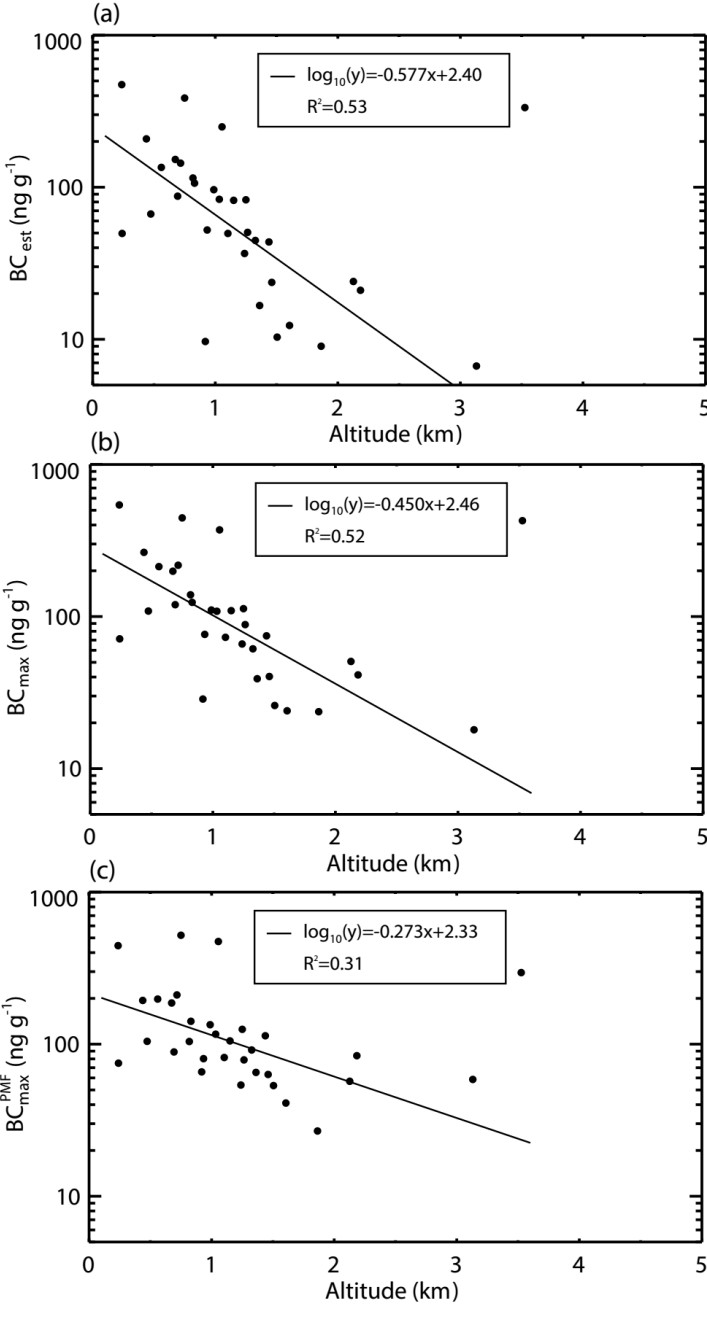

**Fig. 8.** (a) $C_{BC}^{est}$ and (b) $C_{BC}^{max}$ measured according to the ISSW and (C) $C_{BC}^{max}$ determined by the PMF model for the surface snow for each site as a function of altitude for the sites in Xinjiang.



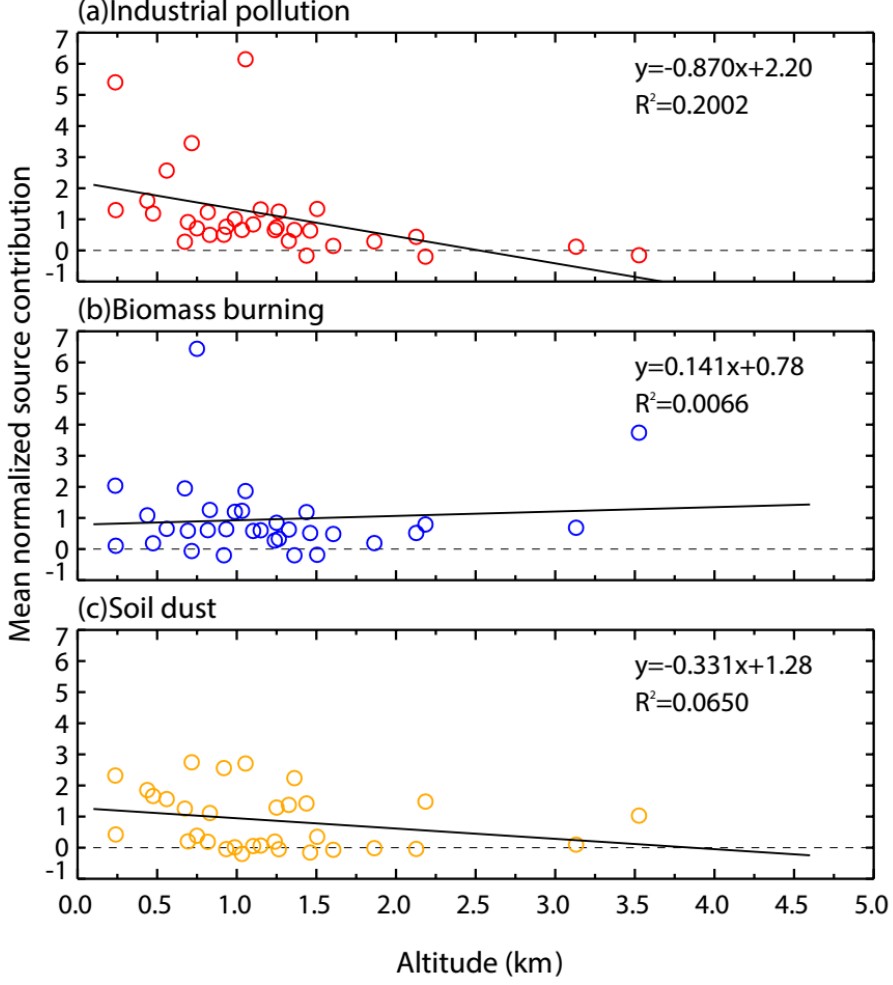

**Fig. 9.** Contributions from each source/factor as a function of altitude for the sites in Xinjiang. The contributions were normalized by the average value of the respective factor contribution over all sites.



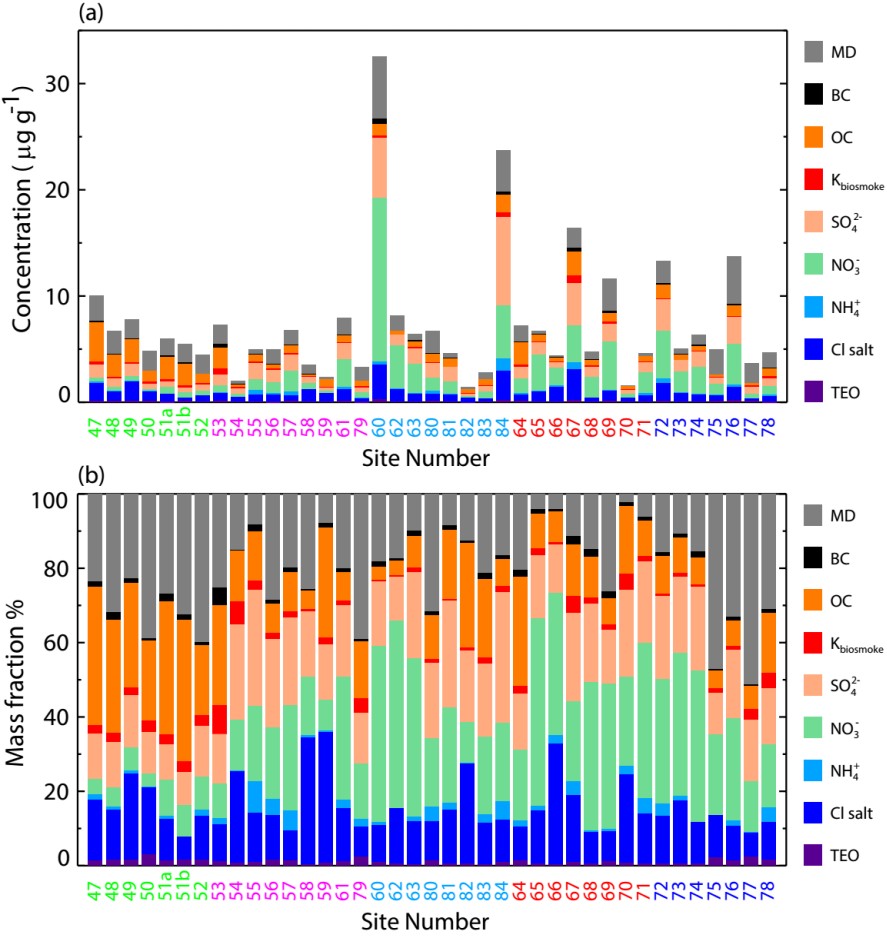

**Fig. 10.** (a) Average mass contributions and (b) average mass fractional contributions of the chemical components in the surface snow at each site.





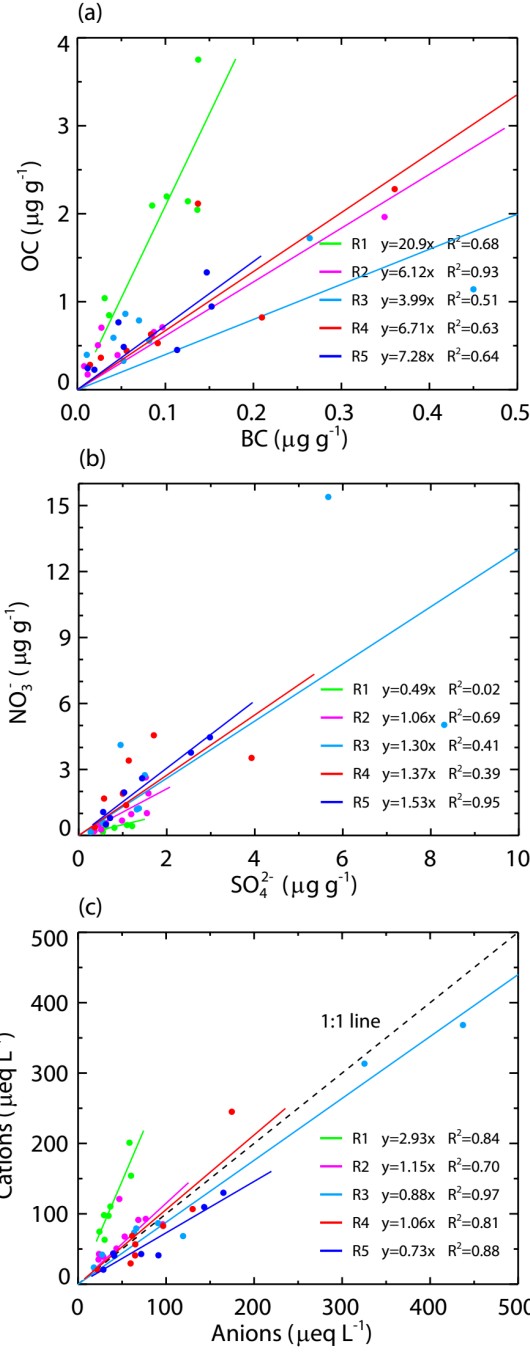

**Fig. 11.** Ratios of the (a) OC and BC mixing ratios and (b) $NO_3^-$ and $SO_4^{2-}$ concentrations, (c) and the charge balance between cations ($Na^+$, $K^+$, $Ca^{2+}$, $Mg^{2+}$ and $NH_4^+$) and anions ($SO_4^{2-}$, $NO_3^-$, $Cl^-$ and $F^-$) for the surface snow samples.



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
