# Peer review of "Properties of black carbon and other insoluble light-absorbing particles in seasonal snow of northwest China"

_The Cryosphere, 2016_

## Referee Comment (RC1) · Anonymous Referee #1 · 21 Dec 2016

Review of "Properties of black carbon and other insoluble light-absorbing particles in seasonal snow of northwest China", by Pu et al

general comments:

This paper deals with light absorbing impurities and their sources in snow in northern China. As the authors clearly state, these impurities impact upon the radiative budget of snow and are therefore important climate agents. These are relevant scientific questions within the scope of TC. This study applies the same general method used in several previous studies (Hegg et al., 2010; Wang et al., 2013; Zhang et al., 2013), and expands geographically on these studies, reaching similar conclusions as to the different sources responsible for those impurities in snow.

[Figure]

The work is generally relevant to the scope of The Cryosphere and is worth publishing, once the authors take care of the following remarks.

My major issues with the manuscript in its current form areĂă:

- General lack of precision and clarity, making the reasoning of the authors very hard to follow.

- Methodological problems with the PMF analysis: Recently, studies have shown the importance of uncertainties evaluation on the results of PMF on aerosols (ex: (Waked et al., 2014)). This lead to the publication of general guidelines for PMF analysis (Belis et al., 2014), that should be adapted here with more clarity. The authors refer to (in particular) (Hegg et al., 2010) for details on the PMF, but as they do not take the same species into account, there is a clear lack of details. In particular, the author seem to be using in their PMF analysis some derived quantities such a Kbiosmoke. How does this concentration depend on assumption on seasalt and crust concentrations ratios ? And does it impact the PMF ?

- Uncertainties analysis: in general, I feel the authors should have done a more thorough uncerty analysis. This is particularly true concerning the retrieval of absorbances from iron oxide, Brown carbon and black carbon from the ISSW measurements. The authors mention (Doherty et al., 2010) and (Grenfell et al., 2011) for error estimates on those measurements, but these references only took into account Black and Brown Carbon, so only partially apply here. See for example the discussion by (Lack and Langridge, 2013)

Specific Comments:

P3 L15: "radiative forcing I highly uncertain": di the authors mean radiative forcing in general, or more precisely in snowy places ? Please precise.

P5 L15: "dust is the main absorber in snow locations" : missing word, many ? Most ?

P6 L1: "quantify the source attribution" → "attribute the sources"

P7 L2: "we evaluated the chemical components to examine the potential emission sources": quite vague

P7 L20-21: how does measuring the snow density and temperature help quantify the deposition flux of BC ?

P8 L7: "nuclepore filters were subjected to BC and OC analyses": this sentence is overly misleading. BC analyses are optical measurements (see (Petzold et al., 2013) for nomenclature), which is actually what is done here. But OC generally refers to carbon measurements made from combustion methods, which is not the case here, and would anyway be impossible on nuclepore filters.

P12 L3-4: "quantify contributions from sources based on composition or fingerprints of the sources" : this seems ill-formulated as actually the PMF gives factors purely from statistical considerations, without any a priori knowledge of eventual "source finger-prints". It is then up to the user to interpret the calculated statistical factors as sources, as the authors actually did

P12 L14-16: from this sentence, the choice of the number of factors seems pretty much to be an arbitrary decision from the user, whereas some "best practices" exist for this choice.

P14 L10-15: the authors mention a potential oulier. Is it the only one ? How were these accounted for in the PMF ?

P20 L15-20: the authors mention "considerable errors": could they be more specific ?

P21 L10-15: it would be good to compare the number of factors to the total number of species taken into accounrt

P21 L16: does really the Figure show "measured mass concentrations" ? Or is it rather calculated masse concentrations (calculated by the PMF)

P23-24: I do not really understand the interest of §3.4.2. As I understand, it discusses

the contribution of a given source to each site, normalized by the average contribution. I do not really see what geochemical information this holds. On the opposite, I understand the following paragraph, where on each site, we have a picture of the origin of LAIs.

P25 L3-10: the authors point that their results differ largely (on the one common region) with previous results from (Zhang et al., 2013), then invoke differences in species taken into account and inconsisntecies in the PMF analysis. This needs to be precised: if results vary too much upon the species taken into account, then there need to be a clear discusion no why your species set is "nest"

P25 L15-20: the correlations showed in figure 9 do not seem very strong. Could the authors give some p-values for those ?

P26 L21: nitrate and sulfate are secondary aerosol, not primary

Belis, C. A., Favez, O., Harrison, R. M., Larsen, B. R., Amato, F., El Haddad, I., Hopke, P. K., Nava, S., Paatero, P., Prévôt, A., Quass, U., et al.: European guide on air pollution source apportionment with receptor models., Publications Office, Luxembourg. [online] Available from: http://dx.publications.europa.eu/10.2788/9307 (Accessed 21 December 2016), 2014.

Doherty, S. J., Warren, S. G., Grenfell, T. C., Clarke, A. D. and Brandt, R. E.: Light-absorbing impurities in Arctic snow, Atmos Chem Phys, 10(23), 11647–11680, doi:10.5194/acp-10-11647-2010, 2010.

Grenfell, T. C., Doherty, S. J., Clarke, A. D. and Warren, S. G.: Light absorption from particulate impurities in snow and ice determined by spectrophotometric analysis of filters, Appl. Opt., 50(14), 2037–2048, doi:10.1364/AO.50.002037, 2011.

Hegg, D. A., Warren, S. G., Grenfell, T. C., Sarah J Doherty and Clarke, A. D.: Sources of light-absorbing aerosol in arctic snow and their seasonal variation, Atmos Chem Phys, 10(22), 10923–10938, doi:10.5194/acp-10-10923-2010, 2010.

Lack, D. A. and Langridge, J. M.: On the attribution of black and brown carbon light absorption using the Ångström exponent, Atmos Chem Phys, 13(20), 10535–10543, doi:10.5194/acp-13-10535-2013, 2013.

Petzold, A., Ogren, J. A., Fiebig, M., Laj, P., Li, S.-M., Baltensperger, U., Holzer-Popp, T., Kinne, S., Pappalardo, G., Sugimoto, N., Wehrli, C., et al.: Recommendations for reporting "black carbon" measurements, Atmos Chem Phys, 13(16), 8365–8379, doi:10.5194/acp-13-8365-2013, 2013.

Waked, A., Favez, O., Alleman, L. Y., Piot, C., Petit, J.-E., Delaunay, T., Verlinden, E., Golly, B., Besombes, J.-L., Jaffrezo, J.-L. and Leoz-Garziandia, E.: Source apportionment of PM10 in a north-western Europe regional urban background site (Lens, France) using positive matrix factorization and including primary biogenic emissions, Atmos Chem Phys, 14(7), 3325–3346, doi:10.5194/acp-14-3325-2014, 2014.

Wang, X., Doherty, S. J. and Huang, J.: Black carbon and other light-absorbing impurities in snow across Northern China: LIGHT-ABSORBING IMPURITIES IN SNOW, J. Geophys. Res. Atmospheres, 118(3), 1471–1492, doi:10.1029/2012JD018291, 2013.

Zhang, R., Hegg, D. A., Huang, J. and Fu, Q.: Source attribution of insoluble light-absorbing particles in seasonal snow across northern China, Atmos Chem Phys, 13(12), 6091–6099, doi:10.5194/acp-13-6091-2013, 2013.
* * *

---

## Referee Comment (RC2) · Anonymous Referee #2 · 30 Dec 2016

I have read the manuscript titled: Properties of black carbon and other insoluble light-absorbing particles in seasonal snow of northwest China. Overall, I feel that the manuscript is well written and is worthy of publication in The Cryosphere. The first referee made multiple comments regarding the chemical analysis. As this is not my area of expertise, I will comment on some additional issues that I noticed.

This research (as well as many publications reporting on this topic) suffers from one common uncertainty. Since results are presented from data collected in one moment in time, how useful are the results in reality? How representative is that one point in time for representing conditions at any other time in the snow year? Different weather conditions can significantly affect the snow ILAP concentrations. While this is a common

problem with this type of measurements that can only be overcome by more intensive sampling, I feel that the authors should include wording that state that the results are from one time measurements at each location and may not be representative of the long term characteristics of the snow in that location.

If I recall correctly, the Hegg technique requires that the snow is relatively fresh. There do not appear to be any comments regarding the time since the most recent snow storm for each of the sites (other than the mention of the 13 sites where it was snowing during collection). Could this affect some of the chemical analysis as some chemical constituents may have washed out of the snowpack?

Page 3 line 11, Using the SNICAR online model (http://snow.engin.umich.edu), using the default snow constants then either 0 or 10 ng/g, the broadband albedo reduction is closer to 0.3% rather than 1%. The Warren and Wiscombe (1980) paper shows possible values with some relatively extreme cases depicted. Could the authors clarify what snow conditions they are using and then clarify the appropriateness of these conditions?

Page 14 line 19: Earlier the authors stated that they sampled in 5 cm steps down through the snow pack. Is this comparable to other studies where you compare the results? If the surface sample is the top 5 cm of snow, how does the density of the snow affect the measurement (if dry deposition on the surface is the main source, then the 5 cm of snow would dilute the measurement significantly with density as an additional factor). If all of the BC is on the surface, then sampling the top 1 cm of snow versus the top 5 cm of snow (assuming uniform density) would give you a factor of five difference in mass mixing ratio since BC is generally reported in a mass per volume unit.

Nomenclature: The different variable names are not well defined at their first use in the manuscript. Ie. What specifically do "MAX", "EQUIV", and "EST" in superscript mean next to C subscript BC? It might be nice to have these and the many others in a table

for easy reference. If they are equivalent to something used in other literature (eBC in Grenfell), please list the equivalents as well.

Page 18 line 2: Something is missing: "vertical differences were missing at could sites, . . ."

---

## Editor Comment (EC1) · F. Dominé (Editor) · 17 Jan 2017

Dear Authors,

Both reviewers have an overall positive evaluation of your paper, but both also note significant deficiencies in your work, which may cast doubt on the robustness of your methods and conclusion. Reviewer 1 in particular is not fully convinced by your use of the PMF method. He also notes a lack of detail or erroneous points in your description of analytical methods. Reviewer 2 questions the time-representativity of your data set. He also makes an interesting comment regarding the radiative impact of BC in snow.

I encourage you to submit a very carefully revised version, where all measurement

uncertainties and data representativity are clearly detailed and their effects taken into account, in particular regarding their possible impact on the conclusions drawn from your PMF analysis. Your modifications will be evaluated critically before possible acceptance of your paper in The Cryosphere.

Best regards

Florent Domine

---

## Author Comment (AC1) · 11 Feb 2017

Response to Referee #1

We are very grateful for the referee's critical comments and suggestions, which have helped us improve the paper quality substantially. We have addressed all of the comments carefully as detailed below in our point-by-point responses. Our responses start with "R:".

Due to all of the formulas and special characters in our responses can't be added in text perform for the submission of interactive comments, we suggested that it should be better to look through the responses by the corresponding PDF files.

[Figure]

general comments: This paper deals with light absorbing impurities and their sources in snow in northern China. As the authors clearly state, these impurities impact upon the radiative budget of snow and are therefore important climate agents. These are relevant scientific questions within the scope of TC. This study applies the same general method used in several previous studies (Hegg et al., 2010; Wang et al., 2013; Zhang et al., 2013), and expands geographically on these studies, reaching similar conclusions as to the different sources responsible for those impurities in snow. The work is generally relevant to the scope of The Cryosphere and is worth publishing, once the authors take care of the following remarks.

R: Thanks for your comments. We have carefully responded the following remarks.

My major issues with the manuscript in its current form are:

(1) General lack of precision and clarity, making the reasoning of the authors very hard to follow.

R: We have revised the whole manuscript to make the manuscript more readable and comprehensible.

(2) Methodological problems with the PMF analysis: Recently, studies have shown the importance of uncertainties evaluation on the results of PMF on aerosols (ex: (Waked et al., 2014)). This lead to the publication of general guidelines for PMF analysis (Belis et al., 2014), that should be adapted here with more clarity. The authors refer to (in particular) (Hegg et al., 2010) for details on the PMF, but as they do not take the same species into account, there is a clear lack of details.

R: Based on the reviewer's comments, a new figure is given to reveal the uncertainty analysis as "Figure S1". We have also added more detailed description about the PMF analysis in section 3.4.1 in the revised manuscript as follow: The concentrations of the components (chemical and optical constituents) and the associated uncertainty datasets were used to run the PMF 5.0 model. General speaking, three to seven fac-

tors, twelve to thirty components and seven or more random seeds were applied with an objective step-by-step methodology in the PMF model to obtain the best solution in accordance with stability, accuracy, performance, and geochemical likeliness. This methodology is a multistep procedure. Firstly, all of the available components are included. Additional adjustments of the selected input components, numbers of factors and random seeds are dependent on an iterative process. We indicate that the choice of uncertainties could lead an important effect on PMF results. Then, we examined several tests for the uncertainties calculations which includes: (1) combining the detection limit (twice of the standard deviation of the blank samples) and the coefficient of variation (standard deviation of repeated analysis divided by mean value of the repeated analysis), which have been performed by Anttila et al. (1995) and Gianini et al. (2012); (2) the uncertainty datasets were calculated by considering the relative uncertainties of the concentration of each component (Waked et al., 2014). We also considered the results of the bootstraps and examinations of the residuals. The ability of the PMF model to replicate experimental concentrations, especially for components regarded as markers of the specific emission sources, is one of the primary principles applied to assess the permanence of the results at each step. More details in PMF model optimization can be found in Waked et al. (2014) and Belis et al. (2014). We used to estimate the fractional contributions to the 650-700-nm particulate absorption by all of the potential emission sources based on two reliable reasons: (1) represents the mass of BC, assuming all of the particulate light absorption (650-700 nm) is related to BC; (2) is only calculated based on the assumed MAC of BC; therefore, the errors of were the lowest among the studied variables. The chemical components were , , , , Na+, K+, , KBiosmoke, Al, Fe, Mn, Cu, Cr and Ba. Finally, a set of high uncertainty datasets was used in this study. For example, the relative uncertainty was 40% for , , and Al, 50 % for , and KBiosmoke, which was comparable with that used in other studies of the spatial variations (Hegg et al., 2009, 2010; Zhang et al., 2013a; Doherty et al., 2014). All these components were described from weak to strong in the PMF on account of their signal-to-noise ratio and the effect on tracing emission sources. The

results indicated that the , „ , Cu and Cr were categorized as "weak". Therefore, the optimal number of factors/sources was four based on the robust and theoretical Q values (Hegg et al., 2009, 2010). However, three-factor provided more physically reasonable results and the most easily identifiable sources, which was consistent with studies of snow in northeastern China (Zhang et al., 2013a) and North America (Doherty et al., 2014). The diagnostic regression R2 value for with this three-factor solution was considerably high (0.87). Hence, we indicated that the three-factor solution was the best choice in this study."

(3) In particular, the authors seem to be using in their PMF analysis some derived quantities such a Kbiosmoke. How does this concentration depend on assumption on seasalt and crust concentrations ratios ? And does it impact the PMF ?

R: The method to derive KBiosmoke concentrations has been described in section 2.2 and have been investigated in previous study (Hsu et al., 2009). For instance, the KBiosmoke as a well-known indicator plays an important role in tracing biomass burning emissions, which has already been widely used in the PMF analysis (Hegg et al., 2009, 2010; Zhang et al., 2013a; Zhang et al., 2013b).

(4) Uncertainties analysis: in general, I feel the authors should have done a more thorough uncertainty analysis. This is particularly true concerning the retrieval of absorption from iron oxide, Brown carbon and black carbon from the ISSW measurements. The authors mention (Doherty et al., 2010) and (Grenfell et al., 2011) for error estimates on those measurements, but these references only took into account Black and Brown Carbon, so only partially apply here. See for example the discussion by (Lack and Langridge, 2013)

R: Based on the reviewer's comments, the following contents are added: "Lack and Langridge (2013) indicated that the attribution biases of BC absorption are from +7% to $-22\%$ by using the AAE in the range of $1.1 \pm 0.3$ instead of 1 as the common default. In order to reduce the uncertainty of BrC absorption at 404 nm less than $\pm 100\%$, the

absolute contribution from BrC absorption must be at least 23% (10%) of that from BC for PAS measurements. Significantly, the variation of AAE plays an important role in affecting the light absorption attribution. However, most of the studies only took BC and BrC (or OC) into account, which ignored the effect of mineral dust on light absorption. For instance, Doherty et al. (2010) revealed that the consideration of dust will add the complexity but does not effectively change the results due to negligible fractional light absorption of dust in some areas such as the Arctic. However, Wang et al. (2013) and Zhang et al. (2013a) indicated that the light absorption of mineral dust could not be negligible across northern China. Therefore, in view of the importance of mineral dust and the complexity of the combination of BC, OC and dust in snow, we did a sensitive test on two possible cases that the mixing ratios of BC, OC and Fe are 100 (15) ng g-1, 1000 (150) ng g-1 and 50 (50) ng g-1 with the fractional absorption of 42% (34%), 54% (43%) and 4% (23%) based on our filed measurements. We estimated the relative uncertainty of the attributed absorption assuming the AAEs of BC, OC and Fe are $1.1\pm0.3$, $6\pm2$ and $3\pm1$ (Doherty et al., 2010; Lack and Langridge, 2013) instead of 1.1, 6 and 3. As shown in Figure S1, the relative uncertainties of BC and OC are from -53% to 29% and -25% to 43%, respectively, which mainly resulted from the variations of AAEs of OC and BC (left panel). The variation of AAEs of Fe has a slight effect on the light absorption. For Fe, the relative uncertainty is from -18% to 22% based on the variation of AAEs. In case 2 (right panel), the fractional absorption of Fe is much more important compared with that in case 1. The relative uncertainties of BC and OC increased and range from -65% to 35% and -40% to 61%, which highlight the varied AAEs of Fe in uncertainty analysis. The analysis indicates that the changes of AAEs of Fe on uncertainty estimates are dependent on the fractional absorption of Fe."

Specific Comments:

P3 L15: "radiative forcing is highly uncertain": did the authors mean radiative forcing in general, or more precisely in snowy places ? Please precise.

R: The sentence has been revised as "However, the regional and global radiative forcing affected by the ILAPs in snow/ice is highly uncertain".

P5 L15: "dust is the main absorber in snow locations" : missing word, many ? Most ?

R: The sentence has been revised as "Recent studies have indicated that the light absorption by mineral dust is mainly related to iron oxides such as goethite and hematite (Alfaro et al., 2004; Lafon et al., 2004, 2006). Although its ability to reduce snow albedo is less than that of BC by approximately a factor of 50 (Warren, 1984). We note that the mass loading of mineral dust could be dominated in several snow sampling locations (Wang et al., 2013)."

P6 L1: "quantify the source attribution" → "attribute the sources"

R: "quantify the source attribution" has been revised as "attribute the sources".

P7 L2: "we evaluated the chemical components to examine the potential emission sources": quite vague

R: The sentence has been revised as "The chemical components and ILAPs were also used to estimated the potential emission sources and source attributions of ILAPs in seasonal snow."

P7 L20-21: how does measure the snow density and temperature help quantify the deposition flux of BC?

R: We are sorry for the misleading. The sentence has been rewritten as "Snow density and snow temperature were also measured within each layer, which could be useful for the parameterization of snow albedo modeling (Flanner et al., 2007, Wang et al., 2017). "

P8 L7: "nuclepore filters were subjected to BC and OC analyses": this sentence is overly misleading. BC analyses are optical measurements (see (Petzold et al., 2013) for nomenclature), which is actually what is done here. But OC generally refers to carbon measurements made from combustion methods, which is not the case here,

and would anyway be impossible on nuclepore filters.

R: We have revised the sentence as "the filters were used to measure the light absorption of ILAPs". The details in separating the light absorption of ILAPs in snow could be found in Wang et al. (2013, 2017).

P12 L3-4: "quantify contributions from sources based on composition or fingerprints of the sources" : this seems ill-formulated as actually the PMF gives factors purely from statistical considerations, without any a priori knowledge of eventual "source fingerprints". It is then up to the user to interpret the calculated statistical factors as sources, as the authors actually did

R: We have revised the sentence as "provides source attribution by identifying and quantifying source profiles and contributions prior, which is based on mathematical approaches".

P12 L14-16: from this sentence, the choice of the number of factors seems pretty much to be an arbitrary decision from the user, whereas some "best practices" exist for this choice.

R: The sentence has been revised as "therefore, the number of factors is unknown priori, which must be selected individually in terms of stability, accuracy, performance, and geochemical likeliness of the PMF results and the analyst's understanding of the sources".

P14 L10-15: the authors mention a potential oulier. Is it the only one ? How were these accounted for in the PMF ?

R: Sorry for the misleading. If we only considered all of the values of during this snow field campaign, the bottom value of at site 83 should be considered as a potential oulier. Then, a possible explanation was given as "however, the underlying soil may have been responsible for this high value. Therefore, we note that this value should not be used to represent the regional background level of BC.". We indicated that only the

chemical species and in surface snow were used as the input parameters for the PMF model. Therefore, the highest value of at the bottom layer at site 83 was irrelevant with the PMF analysis. Then, the following sentence is also revised as "After excluding the bottom value of at site 83".

P20 L15-20: the authors mention "considerable errors": could they be more specific ?

R: We plotted a new figure as Figure S1 to analyze the uncertainties in attributed absorption of BC, OC and Fe at 450 nm associated with the changes of Absorption Angstrom Exponent (AAE). More details could be found in comment (4) and Figure S1.

P21 L10-15: it would be good to compare the number of factors to the total number of species taken into account

R: We have compared the number of factors to the total number of components taken into account, and presented a more detailed description on PMF analysis. More details could be found in comment (2).

P21 L16: does really the Figure show "measured mass concentrations" ? Or is it rather calculated mass concentrations (calculated by the PMF)

R: We have revised "measured mass concentrations" as "calculated mass concentrations".

P23-24: I do not really understand the interest of §3.4.2. As I understand, it discusses the contribution of a given source to each site, normalized by the average contribution. I do not really see what geochemical information this holds. On the opposite, I understand the following paragraph, where on each site, we have a picture of the origin of LAIs.

R: Based on the reviewer's comment, we combine 3.4.2&3.4.3 as section 3.4.3 of source attribution in snow.

P25 L3-10: the authors point that their results differ largely (on the one common region)

with previous results from (Zhang et al., 2013), then invoke differences in species taken into account and inconsisntecies in the PMF analysis. This needs to be precised: if results vary too much upon the species taken into account, then there need to be a clear discussion no why your species set is "nest"

R: We have revised the sentence as "The PMF results in Qinghai in this study were not comparable well with those by Zhang et al. (2013a), who indicated that soil dust was the dominant source of ILAPs. However, the discrepancy could be concluded as (1) the receptor sites between two field campaigns were really far; (2) the chemical components inputs were different (e.g. and KBiosmoke instead of levoglucosan as the markers for biomass burning emissions); (3) the variables that characterized the particulate light absorption in the PMF analysis were inconsistent that we used instead of ILAPs to estimate the fractional contributions to the 650-700-nm particulate absorption.". In addition, we have added a more detailed description on PMF analysis. More details could be found in comment (2).

P25 L15-20: the correlations showed in figure 9 do not seem very strong. Could the authors give some p-values for those ?

R: Based on the reviewer's comments, the Figure 9 was replotted with the confidence level added (See Figure 9). As shown in Fig. 9a, the contributions from industrial pollution sources influenced by human activities is highly related to the altitude, while the biomass burning and soil dust only show weak correlations.

P26 L21: nitrate and sulfate are secondary aerosol, not primary

R: We have revised "primary" as "major".

Fig. S1. Uncertainty in attributed absorption of BC, OC and Fe at 450 nm associated with the changes of Absorption Angstrom Exponent (AAE). Case 1 & 2 represent two typical conditions that the fractional absorption of Fe are 4% and 23% in seasonal snow during this field campaign, respectively.

[revised manuscript text omitted]

Please also note the supplement to this comment:
http://www.the-cryosphere-discuss.net/tc-2016-233/tc-2016-233-AC1-supplement.pdf

[Figure]

**Fig. 1.** Fig. S1. Uncertainty in attributed absorption of BC, OC and Fe at 450 nm associated with the changes of Absorption Angstrom Exponent (AAE).

[Figure]

**Fig. 2.** Fig. 9. Scaled contributions from each source/factor as a function of altitude at sampling sites in Xinjiang.

[Figure]

---

## Author Comment (AC2) · 11 Feb 2017

Response to Referee #2

We are very grateful for the referee's critical comments and suggestions, which have helped us improve the paper quality substantially. We have addressed all of the comments carefully as detailed below in our point-by-point responses. Our responses start with "R:".

Due to all of the formulas and special characters in our responses can't be added in text perform for the submission of interactive comments, we suggested that it should be better to look through the responses by the corresponding PDF files.

[Figure]

I have read the manuscript titled: Properties of black carbon and other insoluble light absorbing particles in seasonal snow of northwest China. Overall, I feel that the manuscript is well written and is worthy of publication in The Cryosphere. The first referee made multiple comments regarding the chemical analysis. As this is not my area of expertise, I will comment on some additional issues that I noticed.

R: Tanks very much for your comments and suggestions, we have addressed all of the comments carefully as detailed below.

This research (as well as many publications reporting on this topic) suffers from one common uncertainty. Since results are presented from data collected in one moment in time, how useful are the results in reality? How representative is that one point in time for representing conditions at any other time in the snow year? Different weather conditions can significantly affect the snow ILAP concentrations. While this is a common problem with this type of measurements that can only be overcome by more intensive sampling, I feel that the authors should include wording that state that the results are from one time measurements at each location and may not be representative of the long term characteristics of the snow in that location.

R: Generally, the sampling sites were selected 50 km away from cities and at least 1 km upwind of the approach road or railway to minimize the effect of pollution from local sources across northwestern China. But we also agree with the reviewer that understanding spatial and temporal differences of ILAPs in snow is still challenges. Therefore, the comparison of the seasonal and interannual variability of the ILAPs in snow was investigated, and the result shows that the differences of ILAPs in snow are relatively small in the Arctic and northeastern China (Doherty et al., 2010; Wang et al., 2013, 2017). We note that further snow field campaigns were still performed worldwide to limit the uncertainties of ILAPs in snow due to the spatial and temporal differences across northern China, the Himalayas, North America, Greenland and the Arctic since 1980s (Cong et al., 2015; Dang and Hegg, 2014; Doherty et al., 2010, 2014; Hegg et al., 2009, 2010; Huang et al., 2011; Xu et al., 2009, 2012; Zhao et al., 2014; Warren

and Wiscombe, 1980, 1985). For instance, a similar paper on the mixing ratios of ILAPs in Arctic snow has been widely used for validating modeled snow BC mixing ratios (Doherty et al., 2010). Therefore, we indicated that the datasets in this study can contribute to advancing remote sensing techniques and reducing the uncertainties of the model simulations to enhance our further understanding ofÂăthe climate impacts of ILAPs in snow and ice.

If I recall correctly, the Hegg technique requires that the snow is relatively fresh.

R: We have carefully looked through the papers by Hegg et al. (2009, 2010). The major points in the literatures are the attribution of the chemical species and the mixing ratios of BC in seasonal snow, which were based on the results of light absorption of BC in both fresh and aged snow by Doherty et al. (2010). Recent studies also mentioned that using the chemical and optical data, which are included both fresh and aged snow samples, were input to a Positive Matrix Factorization (PMF) analysis of the sources of particulate light absorption (Doherty et al, 2014; Zhang et al., 2013). Therefore, the PMF technique is mainly based on different species as the input without considering the snow samples as fresh or aged.

There do not appear to be any comments regarding the time since the most recent snow storm for each of the sites (other than the mention of the 13 sites where it was snowing during collection). Could this affect some of the chemical analysis as some chemical constituents may have washed out of the snowpack?

R: We note that the snow field campaign was conducted in winter season and the surface snow kept frozen and hadn't yet melted during our sampling processes in most of the snow sampling sites. Therefore, the melting or washing effects were negligible in this study. But we also agree with the reviewer that the melting and washed out effect of the snowpack should be considered if the snow samples were really melted. Details for the melting and washed out processes of ILAPs in snow could be found by Wang et al., (2013, Equation 1 & Figure 3) and Doherty et al. (2013, Figure 1) .

Page 3 line 11, Using the SNICAR online model (http://snow.engin.umich.edu), using the default snow constants then either 0 or 10 ng/g, the broadband albedo reduction is closer to 0.3% rather than 1%. The Warren and Wiscombe (1980) paper shows possible values with some relatively extreme cases depicted. Could the authors clarify what snow conditions they are using and then clarify the appropriateness of these conditions?

R: We have revised the sentence as "Warren and Wiscombe (1980) indicated that a mixing ratio of 10 ng g-1 of BC in snow with snow grain size of 1000 $\mu$m may reduce the snow albedo at 400 nm by approximately 1%."

Page 14 line 19: Earlier the authors stated that they sampled in 5 cm steps down through the snow pack. Is this comparable to other studies where you compare the results?

R: Sure, the methods on snow collections and spectrophotometric analysis are definitely comparable with the previous studies (e.g. Doherty et al., 2010, 2014; Wang et al., 2013).

If the surface sample is the top 5 cm of snow, how does the density of the snow affect the measurement (if dry deposition on the surface is the main source, then the 5 cm of snow would dilute the measurement significantly with density as an additional factor).

R: We agree with the reviewer that it is still a challenge to separate the dry and wet deposition of the ILAPs in the top 5 cm snow. The most important reason is that the snow albedo is mainly influenced by the surface snow, especially for the top 5 cm. Therefore, in order to compare and improve the model simulation, we use the top 5 cm as the cumulative values, which include both wet and dry deposition. However, we also indicated the dirty layers and new fallen snow were collected in all sites separately, even for the filtration processes. Doherty et al. (2010) also indicated that if there was obvious layering, for example a thin top layer of newly fallen snow or drift snow, that layer was collected separately, however thin.

If all of the BC is on the surface, then sampling the top 1 cm of snow versus the top 5 cm of snow (assuming uniform density) would give you a factor of five difference in mass mixing ratio since BC is generally reported in a mass per volume unit.

R: The same as above comment. Generally, we collect the vertical profiles of snow sample in every 5 cm when the snow looks uniform. Otherwise, the snow samples will be gathered separately, if it is a significant dirty layer or new fallen snow, whatever 1 cm or thick layers.

Nomenclature: The different variable names are not well defined at their first use in the manuscript. Ie. What specifically do "MAX", "EQUIV", and "EST" in superscript mean next to C subscript BC? It might be nice to have these and the many others in a table for easy reference. If they are equivalent to something used in other literature (eBC in Grenfell), please list the equivalents as well.

R: Thanks for your suggestion. We have added the description of variables in Table 1.

Page 18 line 2: Something is missing: "vertical differences were missing at could sites, . . ."

R: We have revised the sentence as ". . .although apparent vertical differences could be observed at could sites, such as site 47".

References:

Cong, Z., Kang, S., Kawamura, K., Liu, B., Wan, X., Wang, Z., Gao, S., and Fu, P.: Carbonaceous aerosols on the south edge of the Tibetan Plateau: concentrations, seasonality and sources, Atmos. Chem. Phys., 15, 1573-1584, 2015.

Dang, C., and Hegg, D. A.: Quantifying light absorption by organic carbon in Western North American snow by serial chemical extractions, J. Geophys. Res.-Atmos., 119, 247-210, 2014.

Doherty, S. J., Dang, C., Hegg, D. A., Zhang, R., and Warren, S. G.: Black carbon and

other light-absorbing particles in snow of central North America, J. Geophys. Res.-Atmos., 119, 12807-12831, 2014.

Doherty, S. J., Grenfell, T. C., Forsstrom, S., Hegg, D. L., Brandt, R. E., and Warren, S. G.: Observed vertical redistribution of black carbon and other insoluble light-absorbing particles in melting snow, J. Geophys. Res.-Atmos., 118, 5553-5569, 2013.

Doherty, S. J., Warren, S. G., Grenfell, T. C., Clarke, A. D., and Brandt, R. E.: Light-absorbing impurities in Arctic snow, Atmos. Chem. Phys., 10, 11647-11680, 2010.

Hegg, D. A., Warren, S. G., Grenfell, T. C., Doherty, S. J., and Clarke, A. D.: Sources of light-absorbing aerosol in arctic snow and their seasonal variation, Atmos. Chem. Phys., 10, 10923-10938, 2010.

Hegg, D. A., Warren, S. G., Grenfell, T. C., Doherty, S. J., Larson, T. V., and Clarke, A. D.: Source Attribution of Black Carbon in Arctic Snow, Environ. Sci. Technol., 43, 4016-4021, 2009.

Huang, J., Fu, Q., Zhang, W., Wang, X., Zhang, R., Ye, H., and Warren, S. G.: Dust and Black Carbon in Seasonal Snow Across Northern China, Bull. Amer. Meteor. Soc., 92, 175-181, 2011.

Wang, X., Doherty, S. J., and Huang, J.: Black carbon and other light-absorbing impurities in snow across Northern China, J. Geophys. Res.-Atmos., 118, 1471-1492, 2013.

Wang, X., Pu, W., Ren, Y., Zhang, X., Zhang, X., Shi, J., Jin, H., Dai, M., and Chen, Q.: Observations and model simulations of snow albedo reduction in seasonal snow due to insoluble light-absorbing particles during 2014 Chinese survey, Atmos. Chem. Phys., 2017.

Warren, S. G., and Wiscombe, W. J.: A Model for the Spectral Albedo of Snow .2. Snow Containing Atmospheric Aerosols, J. Atmos. Sci., 37, 2734-2745, 1980.

[Figure]

Warren, S. G. and Wiscombe, W. J.: Dirty Snow after Nuclear-War, Nature, 313, 467-470, 1985.

Xu, B. Q., Cao, J. J., Hansen, J., Yao, T. D., Joswia, D. R., Wang, N. L., Wu, G. J., Wang, M., Zhao, H. B., Yang, W., Liu, X. Q., and He, J. Q.: Black soot and the survival of Tibetan glaciers, Proc. Nat. Acad. Sci. U.S.A., 106, 22114-22118, 2009.

Xu, B. Q., Cao, J. J., Joswiak, D. R., Liu, X. Q., Zhao, H. B., and He, J. Q.: Post-depositional enrichment of black soot in snow-pack and accelerated melting of Tibetan glaciers, Environ. Res. Lett., 7, 014022, 2012.

Zhang, R., Hegg, D. A., Huang, J., and Fu, Q.: Source attribution of insoluble light-absorbing particles in seasonal snow across northern China, Atmos. Chem. Phys., 13, 6091-6099, 2013.

Zhao, C., Hu, Z., Qian, Y., Leung, L. R., Huang, J., Huang, M., Jin, J., Flanner, M. G., Zhang, R., Wang, H., Yan, H., Lu, Z., and Streets, D. G.: Simulating black carbon and dust and their radiative forcing in seasonal snow: a case study over North China with field campaign measurements, Atmos. Chem. Phys., 14, 11475-11491, 2014.

Please also note the supplement to this comment:
http://www.the-cryosphere-discuss.net/tc-2016-233/tc-2016-233-AC2-supplement.pdf
* * *
**Table 1.** Variables derived by using the ISSW spectrophotometer.

| Symbols | Description of variables |
|---|---|
| $C_{BC}^{max}$ | Maximum BC is the mass of BC per mass of snow, if all particulate light absorption (650–700nm) is due to BC. |
| $C_{BC}^{est}$ | Estimated BC is the estimated true mass of BC per mass of snow, derived by separating the spectrally resolved total light absorption. |
| $C_{BC}^{equiv}$ | Equivalent BC is the amount of BC that would need to be present in the snow to account for the wavelength-integrated (300–750nm) total light absorption of down-welling solar radiation by all particulate constituents. |
| $\text{Å}_{tot}$ | Absorption Ångström exponent is calculated between 450 and 600nm, for all particulate deposited on the filter. |
| $f_{nonBC}^{est}$ | Fraction of light absorption by non-BC ILAPs, is the absorption by non-BC particulate constituents, weighted by the down-welling solar flux, then spectrally integrated from 300 to 750nm. |

**Fig. 1.** Table 1. Variables derived by using the ISSW spectrophotometer.

---

## Author Comment (AC3) · 11 Feb 2017

Response to Editor

Dear Authors, Both reviewers have an overall positive evaluation of your paper, but both also note significant deficiencies in your work, which may cast doubt on the robustness of your methods and conclusion. Reviewer 1 in particular is not fully convinced by your use of the PMF method. He also notes a lack of detail or erroneous points in your description of analytical methods. Reviewer 2 questions the time-representativity of your data set. He also makes an interesting comment regarding the radiative impact of BC in snow. I encourage you to submit a very carefully revised version, where all measurement uncertainties and data representativity are clearly detailed and their effects taken

into account, in particular regarding their possible impact on the conclusions drawn from your PMF analysis. Your modifications will be evaluated critically before possible acceptance of your paper in The Cryosphere.

Best regards Florent Domine

We are very grateful for the editor's comments and suggestions, which are very helpful for us to improve and clarify the presentation of our results. The following are our key point responses to the editor's comments.

Reviewer 1 in particular is not fully convinced by your use of the PMF method. He also notes a lack of detail or erroneous points in your description of analytical methods.

R: We have added a more detailed description of the PMF method, which includes: (1) A more detailed analysis process; (2) The methods of uncertainties calculations; (3) The principles for selecting the optimal PMF results. We also presented a description of the input components and the values of the associated uncertainties in our study. In addition, the character of each component was identified according the signal-to-noise ratio and the effect on tracing emission sources. Details could be found in the Responses to Referee 1 and the section 3.4.1 in revised manuscript.

Reviewer 2 questions the time-representativity of your data set. He also makes an interesting comment regarding the radiative impact of BC in snow.

R: Generally, the sampling sites were selected 50 km away from cities and at least 1 km upwind of the approach road or railway to minimize the effect of pollution from local sources across northwestern China. But we also agree with the reviewer that understanding spatial and temporal differences of ILAPs in snow is still challenges. Therefore, the comparison of the seasonal and interannual variability of the ILAPs in snow was investigated, and the result shows that the differences of ILAPs in snow are relatively small in the Arctic and northeastern China (Doherty et al., 2010; Wang et al., 2013, 2017). We note that further snow field campaigns were still performed worldwide

to limit the uncertainties of ILAPs in snow due to the spatial and temporal differences across northern China, the Himalayas, North America, Greenland and the Arctic since 1980s (Cong et al., 2015; Dang and Hegg, 2014; Doherty et al., 2010, 2014; Hegg et al., 2009, 2010; Huang et al., 2011; Xu et al., 2009, 2012; Zhao et al., 2014; Warren and Wiscombe, 1980, 1985). For instance, a similar paper on the mixing ratios of ILAPs in Arctic snow has been widely used for validating modeled snow BC mixing ratios (Doherty et al., 2010). Therefore, we indicated that the datasets in this study can contribute to advancing remote sensing techniques and reducing the uncertainties of the model simulations to enhance our further understanding ofÂăthe climate impacts of ILAPs in snow and ice.

References:

Cong, Z., Kang, S., Kawamura, K., Liu, B., Wan, X., Wang, Z., Gao, S., and Fu, P.: Carbonaceous aerosols on the south edge of the Tibetan Plateau: concentrations, seasonality and sources, Atmos. Chem. Phys., 15, 1573-1584, 2015.

Dang, C., and Hegg, D. A.: Quantifying light absorption by organic carbon in Western North American snow by serial chemical extractions, J. Geophys. Res.-Atmos., 119, 247-210, 2014.

Doherty, S. J., Dang, C., Hegg, D. A., Zhang, R., and Warren, S. G.: Black carbon and other light-absorbing particles in snow of central North America, J. Geophys. Res.-Atmos., 119, 12807-12831, 2014.

Doherty, S. J., Warren, S. G., Grenfell, T. C., Clarke, A. D., and Brandt, R. E.: Light-absorbing impurities in Arctic snow, Atmos. Chem. Phys., 10, 11647-11680, 2010.

Hegg, D. A., Warren, S. G., Grenfell, T. C., Doherty, S. J., and Clarke, A. D.: Sources of light-absorbing aerosol in arctic snow and their seasonal variation, Atmos. Chem. Phys., 10, 10923-10938, 2010.

Hegg, D. A., Warren, S. G., Grenfell, T. C., Doherty, S. J., Larson, T. V., and Clarke,

A. D.: Source Attribution of Black Carbon in Arctic Snow, Environ. Sci. Technol., 43, 4016-4021, 2009.

Huang, J., Fu, Q., Zhang, W., Wang, X., Zhang, R., Ye, H., and Warren, S. G.: Dust and Black Carbon in Seasonal Snow Across Northern China, Bull. Amer. Meteor. Soc., 92, 175-181, 2011.

Wang, X., Doherty, S. J., and Huang, J.: Black carbon and other light-absorbing impurities in snow across Northern China, J. Geophys. Res.-Atmos., 118, 1471-1492, 2013.

Wang, X., Pu, W., Ren, Y., Zhang, X., Zhang, X., Shi, J., Jin, H., Dai, M., and Chen, Q.: Observations and model simulations of snow albedo reduction in seasonal snow due to insoluble light-absorbing particles during 2014 Chinese survey, Atmos. Chem. Phys., 2017.

Warren, S. G., and Wiscombe, W. J.: A Model for the Spectral Albedo of Snow .2. Snow Containing Atmospheric Aerosols, J. Atmos. Sci., 37, 2734-2745, 1980.

Warren, S. G. and Wiscombe, W. J.: Dirty Snow after Nuclear-War, Nature, 313, 467-470, 1985.

Xu, B. Q., Cao, J. J., Hansen, J., Yao, T. D., Joswia, D. R., Wang, N. L., Wu, G. J., Wang, M., Zhao, H. B., Yang, W., Liu, X. Q., and He, J. Q.: Black soot and the survival of Tibetan glaciers, Proc. Nat. Acad. Sci. U.S.A., 106, 22114-22118, 2009.

Xu, B. Q., Cao, J. J., Joswiak, D. R., Liu, X. Q., Zhao, H. B., and He, J. Q.: Post-depositional enrichment of black soot in snow-pack and accelerated melting of Tibetan glaciers, Environ. Res. Lett., 7, 014022, 2012.

Zhao, C., Hu, Z., Qian, Y., Leung, L. R., Huang, J., Huang, M., Jin, J., Flanner, M. G., Zhang, R., Wang, H., Yan, H., Lu, Z., and Streets, D. G.: Simulating black carbon and dust and their radiative forcing in seasonal snow: a case study over North China with field campaign measurements, Atmos. Chem. Phys., 14, 11475-11491, 2014.

Please also note the supplement to this comment:
http://www.the-cryosphere-discuss.net/tc-2016-233/tc-2016-233-AC3-supplement.pdf
* * *

---

## Author Response (AR2)

**Response to Editor**

Comments to the Author:
Dear Authors

Thank you for your substantially improved revised version. I have read your manuscript carefully and have obtained the evaluation from one of the initial reviewers, who is also satisfied with your changes. I am therefore pleased to accept your paper for publication in The Cryosphere, pending minor language changes as outlined by the reviewer.
Thank you for submitting your work to The Cryosphere.

Florent Domine
Co-Editor-in-Chief

We are very grateful for the editor's comments and suggestions, which are very helpful for us to improve and clarify the presentation of our results. We have carefully made technical corrections suggested by the reviewer.

**Response to Anonymous Referee #1**

We are very grateful for the referee's critical comments and suggestions, which have helped us improve the paper quality substantially. We have addressed all of the comments carefully as detailed below in our point-by-point responses. Our responses start with "R:".

The authors have made a significant effort at clarifying their initial manuscript. Some technical points remain to be addressed:

R: Thanks for your comments. We have carefully responded the following technical corrections.

Some specifics:

p3-L14: « However, the regional and global radiative forcing affected by the ILAPs in snow/ice is still a challenge »: « however, evaluating how much ILAPs in snow/ice affect the regional and global radiative forcing still a challenge »

R: We have revised the sentence as "However, evaluating how much ILAPs in snow/ice affect the regional and global radiative forcing is still a challenge".

p5-L7: « In addition to BC, which presented … »: « In addition to BC, which is... »

R: We have revised the sentence as "In addition to BC, which is...".

p5-L15: « Although its ability to reduce snow albedo is less than that of BC by approximately a factor of 50 (Warren, 1984). We note that the mass loading of mineral dust could be dominated in several snow sampling locations (Wang et al., 2013) »:« Although its ability to reduce snow albedo is less than that of BC by approximately a factor of 50 (Warren, 1984), we note that the mass loading of mineral dust could dominate in several snow sampling locations (Wang et al., 2013) »

R: The sentence has been modified as "Although its ability to reduce snow albedo is less than that of BC by approximately a factor of 50 (Warren, 1984), we note that the mass loading of mineral dust could dominate in several snow sampling locations (Wang et al., 2013)".

p12L9: I suggest adding « in aerosol studies » after « widely applied »

R: We have added "in aerosol studies" after "widely applied".

p16L14: « compared with the values to the remote northeast »: maybe better as « compared with the values in the remote northeast »

R: The sentence has been revised as "compared with the values in the remote northeast".

p16L15: « considerable »: « considerably »

R: "considerable" has been replaced as "considerably".

p16L19: « despite » → « although »

R: "despite" has been replaced as "although".

p17L10: « The fest varied obviously varied from < 50% to > 90% » : one of the varied is too much

R: We have deleted the second "varied".

p17L11: « indicates the spatial variance in the » → « reflects the spatial variability of »

R: The sentence has been revised as "reflects the spatial variability of".

p22L2: "general speaking" --> generally speaking

R: "General speaking" has been replaced as "Generally speaking".

p27L9: « in this study were not comparable well » → « in this study did not compare well »

R: The sentence has been modified as "in this study did not compare well".

[revised manuscript text omitted]